# Targeting neddylation sensitizes colorectal cancer to topoisomerase I inhibitors by inactivating the DCAF13-CRL4 ubiquitin ligase complex

Yilun Sun [1] ✉, Simone A. Baechler[1], Xiaohu Zhang[2], Suresh Kumar[1], Valentina M. Factor[1], Yasuhiro Arakawa[1], Cindy H. Chau [3], Kanako Okamoto[1], Anup Parikh[4], Bob Walker [5], Yijun P. Su[6], Jiji Chen [6], Tabitha Ting[7], Shar-yin N. Huang[1], Erin Beck[2], Zina Itkin[8], Crystal McKnight[8], Changqing Xie [9], Nitin Roper[1], Deepak Nijhawan[6], William Douglas Figg[3], Paul S. Meltzer [5], James C. Yang [4], Craig J. Thomas [2] & Yves Pommier [1] ✉

Colorectal cancers (CRCs) are prevalent worldwide, yet current treatments remain inadequate. Using chemical genetic screens, we identify that co-inhibition of topoisomerase I (TOP1) and NEDD8 is synergistically cytotoxic in human CRC cells. Combination of the TOP1 inhibitor irinotecan or its bioactive metabolite SN38 with the NEDD8-activating enzyme inhibitor pevonedistat exhibits synergy in CRC patient-derived organoids and xenografts. Mechanistically, we show that pevonedistat blocks the ubiquitin/proteasome-dependent repair of TOP1 DNA-protein crosslinks (TOP1-DPCs) induced by TOP1 inhibitors and that the CUL4-RBX1 complex (CRL4) is a prominent ubiquitin ligase acting on TOP1-DPCs for proteasomal degradation upon auto-NEDD8 modification during replication. We identify DCAF13, a DDB1 and Cullin Associated Factor, as the receptor of TOP1-DPCs for CRL4. Our study not only uncovers a replication-coupled ubiquitin-proteasome pathway for the repair of TOP1-DPCs but also provides molecular and translational rationale for combining TOP1 inhibitors and pevonedistat for CRC and other types of cancers.

Colorectal cancer (CRC) remains one of the most frequent malignancies with 177,000 new cases and 58,000 deaths per year in the USA[1], with one-fifth of CRC patients presenting metastatic disease (mCRC). Currently, irinotecan in combination with fluorouracil is the first-line chemotherapy for mCRC[2,3]. Novel molecular targets for combination with irinotecan are needed to improve CRC treatment.

Irinotecan is a prodrug converted within the cell to its active metabolite SN38, a potent camptothecin (CPT)-based drug targeting topoisomerase I (TOP1)[2,4]. TOP1 is nuclear enzyme that relieves DNA torsional stress arising from replication, transcription, and chromatin compaction and relaxation[5,6]. It dissipates DNA supercoils by cleaving one strand of the DNA double helix, allowing the broken strand to rotate (swivel) around the other strand. While cutting the DNA, TOP1 forms a transient catalytic intermediate termed TOP1 cleavage complex (TOP1cc) through a phosphotyrosyl linkage between the catalytic tyrosine residue of TOP1 and the 3' end of the DNA. TOP1ccs are self-reversed upon resealing of the DNA break. CPT analogs trap TOP1ccs

by binding at the interface between the enzyme and the DNA thereby occluding the resealing of the DNA break[7,8]. The resulting irreversible TOP1ccs (which we refer to as TOP1 DNA-protein crosslinks or TOP1-DPCs), if left unrepaired, pose a serious threat to the genome as their bulky protein constituent blocks all chromatin-based processes and most importantly DNA replication[9].

The ability of cancer cells to repair TOP1-DPCs is a key for their resistance to camptothecins[10,11]. A pivotal step is the proteolysis of TOP1-DPCs, which enables tyrosyl-DNA phosphodiesterase 1 (TDP1) to hydrolyze the otherwise concealed phosphotyrosyl bond and endonucleases such as the Mre11-Rad50-Nbs1 complex to excise the DPCs by cleaving the adjacent DNA backbone[12–15]. Although not fully understood, the proteolysis can be catalyzed by the ubiquitin-proteasome pathway (UPP) or by proteases including SPRTN or FAM111A, both of which are activated upon DNA replication collisions[16–19]. Nonetheless, it remains largely unknown whether and how the UPP is activated against TOP1-DPC and regulated. We recently demonstrated that SUMOylation recruits the ubiquitin ligase RNF4 for the ubiquitylation and sequent proteasomal degradation of TOP1-DPCs in a replication- and transcription-independent manner[20]. Because the SUMO pathway does not fully account for TOP1-DPC ubiquitylation and proteasomal degradation, it therefore can be conjectured that a parallel UPP partakes in TOP1-DPC removal in the context of DNA replication.

Neddylation is a post-translational modification by which the ubiquitin-like protein NEDD8 covalently targets substrate proteins through an enzymatic cascade akin to ubiquitylation[21,22]. Neddylation plays a crucial role in cell viability and development and has been implicated in the repair of DNA damage such as DNA double-strand breaks[23–25]. It is yet unknown whether neddylation plays a role in the repair of DNA-protein crosslinks.

In this work, we identify pevonedistat (PEV), a first-in-class inhibitor of neddylation recently approved for the treatment of high-risk myelodysplastic syndromes (HR-MDS), as synergistically cytotoxic with TOP1 inhibitors by high-throughput screens in human colorectal carcinoma cells. Neddylation activates the cullin 4 (CUL4)-RBX ubiquitin ligases (CRL4), which leads to K48-linked polyubiquitylation of TOP1-DPCs and their proteasomal degradation during DNA replication. We also discover that DCAF13, an understudied family member of the DDB1- and CUL4-asscoiated factors, binds the TOP1 core domain on nascent chromatin through its WD40 repeats, linking TOP1-DPCs to DDB1-CRL4 for the DPC ubiquitylation. These findings reveal a salient role of neddylation in the repair of TOP1-DPCs associated with DNA replication. They also provide proof of principle for combining pevonedistat with TOP1 inhibitors to treat CRCs and potentially other types of cancers.

## Results

### High-throughput screening identifies the synergistic combination of pevonedistat and TOP1 inhibitors in CRC cells

To identify effective small molecule inhibitors against CRC, we first assessed 2480 oncology-focused, approved, and investigational drugs in a library termed the NCATS Mechanism Interrogation Plate (MIPE) 5.0[26] in HCT116 human colorectal carcinoma cells. The library exploits redundancy by including multiple inhibitors of well-explored targets while encompassing mechanistic diversity, targeting more than 860 distinct molecular targets. We identified pevonedistat (PEV, also known as MLN-4924), as exhibiting a desired cytotoxicity profile (Fig. 1a and Supplementary Data S1). PEV is an adenosine monophosphate (AMP) mimetic, which forms a stable covalent adduct with NEDD8 in the catalytic pocket of the NEDD8-activating enzyme (NAE) heterodimer by competing with AMP and reacting with thioester-linked NEDD8 bound to the enzyme's catalytic cysteine[27]. This leads to inhibition of neddylation, a ubiquitin-like modification driving important biological processes including cell cycle regulation, viability and

tissue development[21]. PEV is a first-in-class inhibitor currently in multiple clinical trials[28].

We next sought to explore synergistic drug combinations by testing PEV combinations with the entire MIPE 5.0 library in HCT116 cells. PEV-drug pairs were ranked using the Excess over the Highest Single Agent (ExcessHSA) metric to quantitatively assess synergism and antagonism (Fig. 1b). We exploited the mechanistic redundancy built into MIPE 5.0 to generate a pre-ranked drug-target enrichment analysis of the PEV-drug interaction landscape. Synergy with PEV was observed for inhibitors of BRD4 (e.g., mivebresib), glutaminase (e.g., telaglenastat), IAP (e.g., GDC-0152) and TOP1 (e.g., camptothecin, irinotecan, SN38 and topotecan) (Fig. 1c, d and Supplementary Data S2). As irinotecan is used in first- and second-line treatment regimens for metastatic CRC, we centered our subsequent analyses on TOP1 inhibitors and investigated the mechanistic basis of their previously unreported synergy with PEV.

To confirm the synergy of PEV plus TOP1 inhibitors in CRC cells, we tested combinations of PEV with SN38, the active metabolite of irinotecan, by performing the ATPlite luminescence assay to measure cell viability in 6 other cultured CRC cell lines including 4 MSI (Microsatellite Instability)-positive CRC lines SW48, SW837, KM12 and HCT15 (Fig. 1e) and 2 MSI-negative and CIN (chromosome instability)-positive CRC lines HT29 and SW620. All the cell lines exhibited superadditive response to the combination (Supplementary Data Fig. 1a). In addition, we measured cleaved caspase-3, a biomarker for apoptosis by Western blotting (WB) and enzyme-linked immunosorbent assay (ELISA) in CRC cells and found that SN38 in combination PEV induced increased levels of cleaved caspase-3 (Supplementary Data Fig. 1b, c), indicating that the combination not only enhances growth inhibition but also apoptotic cell killing.

As both the combination screen and ATPlite assays showed that PEV did not synergize with TOP2 and PARP inhibitors and other DNA damaging agents (Supplementary Data Fig. 1d), we hypothesized that PEV sensitizes CRC cells to TOP1 inhibitors by specifically inhibiting the repair of TOP1-induced DNA damage. Additional pharmacological analyses using the CellMiner Cross-Database (CellMiner CDB) (http://discover.nci.nih.gov/cellminercdb)[29] showed that the activities of clinical TOP1 inhibitors (irinotecan and topotecan) and PEV are highly correlated across large panels of cancer cell lines (Supplementary Data Fig. 1e), implying TOP1 inhibitors with PEV as a potential therapy for cancers beyond CRC.

### Irinotecan in combination with pevonedistat exhibited synergy in preclinical CRC models

To interrogate the combination beyond cell lines, we employed 3D organoid models derived from three mCRC patients, which closely reproduce the genetic and morphologic heterogeneity of CRC cells in the primary cancer tissue. Following establishment of the three patient-derived organoids (PDOs #1, #2 and #3), we first performed RNA sequencing (RNA-seq) to profile mRNA gene expression in all three organoids and to identify differentially expressed genes. A total of 16,784 genes were detected in the samples, and the RNA-seq analysis showed that colon cancer markers genes were significantly differentially expressed in all the organoids. Specifically, colon cancer marker genes such as ANXA1, FABP6, ACE2, FXYD5, LY6E, SERPINE2, SCD, BMP4, CEACAM6, TESC, and TGFBI were overexpressed in CRC organoids (Supplementary Data Fig. 2a and Supplementary Data S3). Next, we assessed the combination of SN38 with PEV by conducting CellTiter-Glo 3D cell viability assays (Fig. 2a, b). As expected, Synergy of the combination was observed in all three CRC PDOs.

We next assessed the combination of PEV and irinotecan in HCT116 xenograft model[30]. Mice were treated intraperitoneally with 2.5 or 15 mg/kg PEV (5 times weekly, 3 weeks), 20 mg/kg irinotecan (IRI) (QW, 3 weeks), or with the two drugs in combination (Supplementary Data Fig. 2b). While PEV in combination with irinotecan significantly

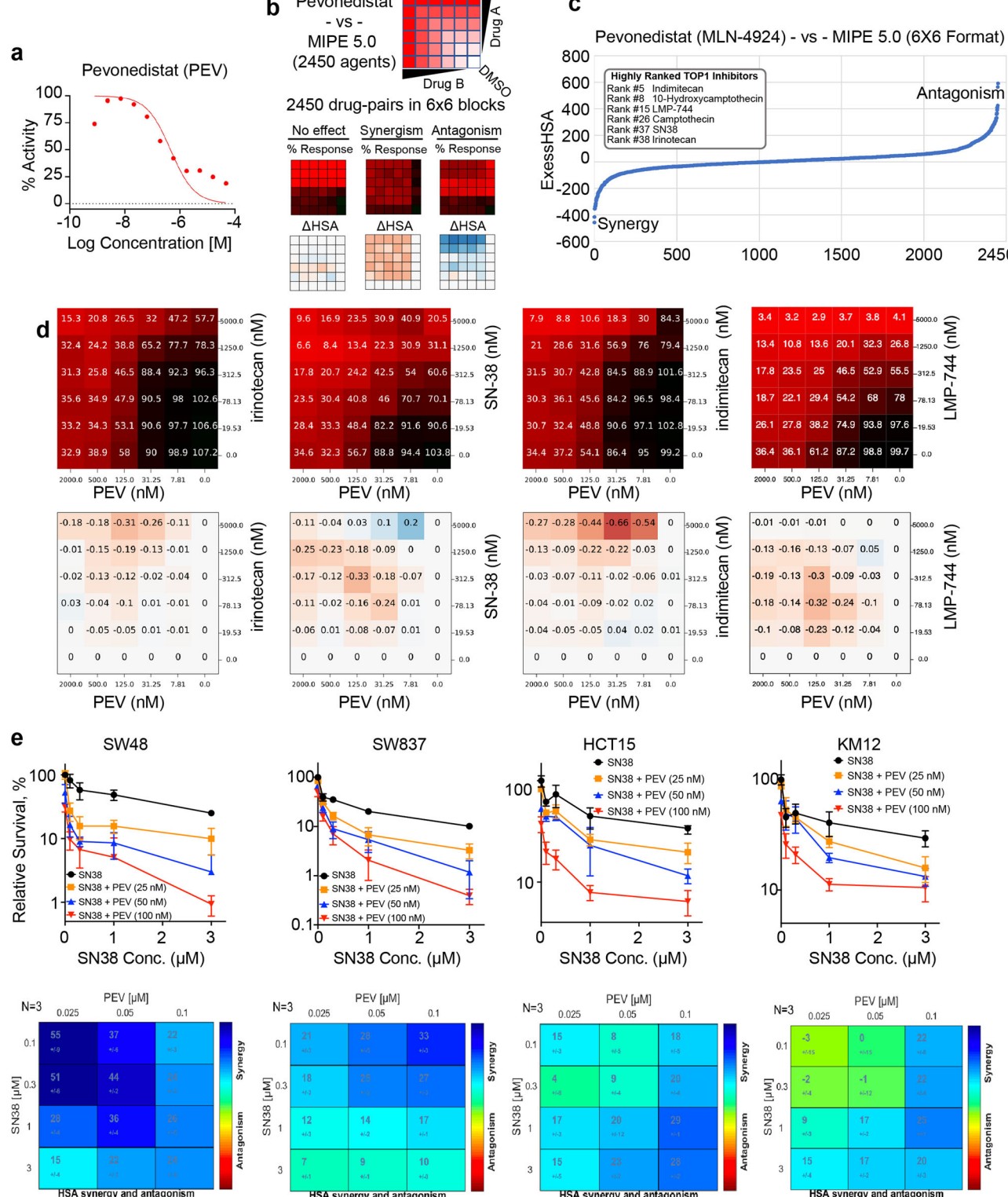

**Fig. 1 | High-throughput screening identifies pevonedistat (PEV) with TOP1 inhibitors as synergistic combination in CRC cells. a** Dose-response curve of PEV derived from MIPE 5.0 library screen in HCT116 CRC cells. **b** Scheme of 6 × 6 matrix screening to examine PEV in combination with the MIPE5.0 library in HCT116 cells. In response matric, red indicates strong response whereas black indicates poor response. In excess over the Highest Single Agent (ExcessHSA or ΔHSA) metric, orange indicates synergy whereas blue indicates resistance. **c** Drug-target enrichment analysis plots highlighting the synergy of PEV with TOP1 inhibitors. PEV-TOP1 inhibitor pairs ranked using the ExcessHSA metric. **d** Response (top panels) and

ΔHSA (bottom panels) heatmaps for the combination of PEV with camptothecins irinotecan and SN38 (the bioactive metabolite of irinotecan) and indenoisoquinolines LMP776 (indimitecan) and LM744 across defined concentration ranges in HCT116 cells. **e** Top panels: viability curves for 72 h treatments with SN38 at defined concentrations in the indicated CRC cell lines (mean ± SD, N = 3 biologically independent experiments) using ATPlite Luminescence Assay. Cells were treated with PEV at defined concentrations 4 h before SN38. Bottom panels: SN38-PEV pairs ranked by ExcessHSA metric using Combenefit, an interactive platform for the analysis of drug combinations.

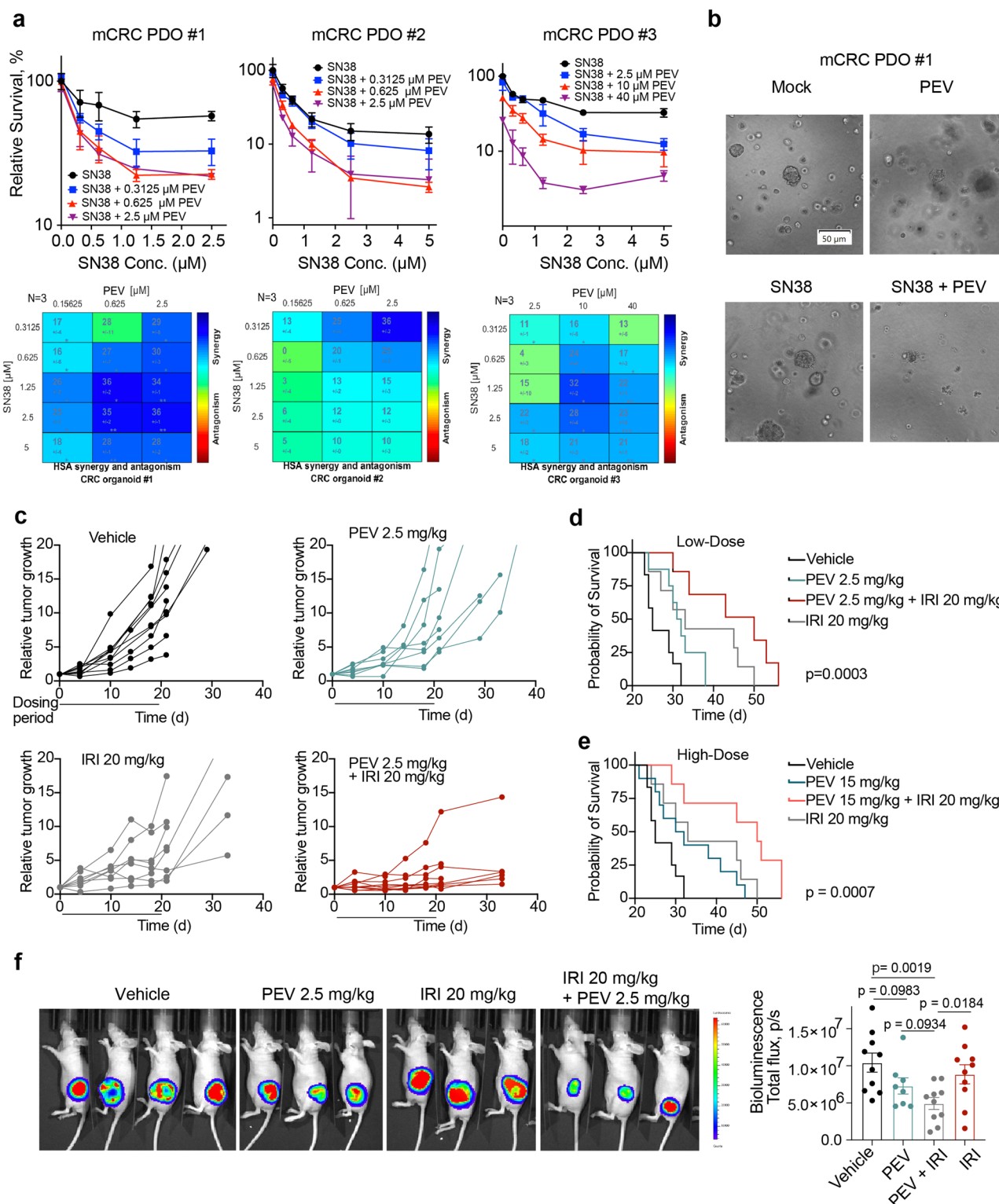

slowed tumor growth and decreased tumor mass, neither of the monotherapies reduced tumor volume and mass significantly compared with the vehicle group (Fig. 2c and Supplementary Data Fig. 2c, d). No toxicity was observed with the combination therapy, as assessed by weight loss (Supplementary Data Fig. 2e, f), clinical chemistry including liver enzymes and renal function biomarkers, complete blood count (CBC), neutrophil count (Supplementary Data Fig. 2g), as well as gross lesions including reactive lymph nodes, reactive gut associated lymphoid tissue, changes in the lung consistent with agonal changes, and changes in the reproductive tract throughout the 3-week treatment period (Supplementary Data S4). The efficacy of the combination treatment was further demonstrated by prolonged survival following the dosing period compared with either the vehicle or the monotherapy groups (Fig. 2d). While increased exposure of PEV (15 mg/kg) in the combination treatment did not result in further enhanced survival, the higher dose extended the survival rate in the monotherapy group (Fig. 2e). Bioluminescent imaging on day 21 after the start of treatment showed a significant reduction in tumor burden in animals treated with the combination therapy compared with mice treated with irinotecan or the vehicle (Fig. 2f).

**Fig. 2 | Irinotecan in combination with PEV exhibits synergy in preclinical CRC models. a** Top panels: Three 3D organoids generated from liver-dominant metastatic CRC patients (mCRC PDOs #1, #2 and #3) were treated with SN38 plus PEV as indicated for 72 h for CellTiter-Glo Luminescent Cell Viability Assay (mean ± SD, $N = 3$ biologically independent experiments). Bottom panels: SN38-PEV pairs ranked by ExcessHSA metric using Combenefit. **b** Brightfield images of mCRC PDO #1 treated with mock (no treatment), 2.5 μM PEV, 1.25 μM SN38 or 2.5 μM PEV + 1.25 μM SN38. The scale bar represents 50 μm. **c** HCT116 tumor-bearing athymic nude mice were treated either with vehicle, PEV (2.5 mg/kg), irinotecan (20 mg/kg) or a combination of PEV plus irinotecan. Treatments were initiated 10 days after tumor inoculation when mean tumor size reached 95.7 mm³. Tumor growth is shown as fold-change versus first day of treatment ($n = 10$ per treatment group, except $n = 8$ for PEV 2.5 mg/kg). **d** Kaplan−Meier survival curve of mice following the three treatment cycles ($n = 10$ in vehicle and PEV group, $n = 7$ all other groups). Survival after treatment with low dose PEV (2.5 mg/kg) in combination with irinotecan. $p$ value determined by Mantel−Cox test. **e** Kaplan−Meier survival curve of mice following the three treatment cycles ($n = 10$ in vehicle and PEV group, $n = 7$ all other groups). Survival after treatment with high dose (15 mg/kg) in combination with irinotecan. $p$ value determined by Mantel−Cox test. **f** Left panel: representative bioluminescence images 21 days after treatment start; Right panel: Quantification of bioluminescence imaging ($n = 10$ per treatment group, except $n = 8$ for PEV 2.5 mg/kg). Data are shown as mean ± SEM. $p$ values determined by two-tailed Student's $t$ test.

Additional assessment of antitumor activity of a combination treatment of PEV with another topoisomerase I inhibitor, topotecan, showed similar results. Treatment of HCT116 xenografts with PEV (15 mg/kg, 5x per week for 3 weeks) together with topotecan (5 mg/kg, twice weekly for 3 weeks) resulted in a significant reduction in tumor growth and extended survival rate (Supplementary Data Fig. 2h, i). In line with the results of the combination treatment with irinotecan, no toxicity was observed with the combination treatment of PEV with topotecan (Supplementary Data Fig. 2j). Notably, topotecan in combination with PEV significantly reduced the tumor mass as quantitated by bioluminescence imaging on day 21 after the start of treatment (Supplementary Data Fig. 2k). These results demonstrate the beneficial effect of PEV in combination with TOP1 inhibitors in CRC preclinical models.

As irinotecan is given in combination with folinic acid (leucovorin) and fluorouracil (5-FU) as the combination regimen (FOLFIRI) for treatment of mCRC, we next evaluated the combination of FOLFIRI with PEV in CRC PDO #2 and observed increased sensitivity to FOLFIRI upon addition of PEV (Supplementary Data Fig. 2l), suggesting the potential of combining FOLFIRI with PEV in clinical trials.

## Neddylation fosters the repair of TOP1-DPCs and activates replication-associated DNA damage responses

Generation of TOP1-DPCs is the therapeutic mechanism of camptothecin- and indenoisoquinoline-based TOP1 inhibitors[4,20]. To interrogate whether PEV sensitizes CRC cells to TOP1 inhibitors by enhancing TOP1-DPCs, we performed in vivo complex of enzyme (ICE) assays[31] (Fig. 3a). While the levels of TOP1-DPCs decreased after 4-h SN38 treatment in HCT116 cells without PEV pre-treatment, cells pre-treated with PEV displayed a significant delay in the clearance of TOP1-DPCs at 4 h (Fig. 3b). Similarly, we observed in our CRC PDO #1 as well as CRC xenografts that PEV inhibited the removal of SN38-induced TOP1-DPCs (Supplementary Data Fig. 3a, b). These results are in line with our finding that TOP1-DPCs peak 30 min after CPT treatment and start to be degraded and removed by the UPP within 1 h of CPT treatment[20]. Considering that p53-negative CRC cells exhibited hypersensitivity to PEV[32], we next sought to determine whether p53 affects the neddylation-dependent repair of TOP1-DPCs. By performing ICE assays, we found that depletion of p53 in SN38-treated HCT116 cells did not impact the levels of TOP1-DPCs and TOP1-DPC accumulation by PEV (Supplementary Data Fig. 3c), excluding the involvement of p53 in the repair of TOP1-DPCs.

To assess how neddylation acts in the UPP, we employed single-molecule fluorescence microscopy to monitor TOP1-HaloTag in live HCT116 cells. Consistent with our recent study in U2OS cells[14], we observed populations of TOP1 single molecules with different dynamics (jump distances ranged from in 0 to 1.2 μm) in HCT116 cells treated with DMSO, the drug vehicle (Fig. 3c middle and bottom panels and Supplementary Movie 1). Upon exposure to SN38 for 2 h, jump distances of the majority of TOP1 single molecules was significantly reduced to 0 to 0.1 μm, consistent with the trapping of TOP1ccs (TOP1-DPC formation). SN38 also decreased the number of TOP1 single molecules (Fig. 3c middle and bottom panels and Supplementary Movie 2), reflecting the downregulation of TOP1 upon TOP1-DPC induction as shown in Fig. 3b[33]. Yet, pre-treatment with PEV followed by co-treatment with SN38 in part prevented SN38-induced TOP1 downregulation without affecting SN38-reduced mobility of TOP1 as did co-treatment with SN38 plus the ubiquitin-activating enzyme (UAE) inhibitor TAK243 (TAK) or the proteasome inhibitor bortezomib (BTZ)[14,20] (Fig. 3c middle and bottom panels and Supplementary Movie 3−5). Of note, pre-treatment with PEV + TAK243 or BTZ did not further increase TOP1 single molecules in comparison with the respective single pre-treatment (Fig. 3c middle and bottom panels and Supplementary Movie 6−8), suggesting that neddylation and the UPP are epistatic for TOP1-DPC removal. This conclusion was further consolidated by our ICE and immunofluorescence assays showing that PEV + TAK243 or BTZ did not further enhance SN38-induced TOP1-DPCs (Fig. 3d and Supplementary Data Fig. 3d, e) and by Western blotting assays showing that PEV + TAK243 or BTZ restored SN38-induced cellular TOP1 downregulation but did not further increase its overall levels (Fig. 3e). Together, these results demonstrate that PEV, by inhibiting neddylation, blocks the repair of TOP1-DPCs.

Collisions between replication forks and TOP1-DPCs results in single-ended DNA double-strand breaks (seDSBs) by replication run-off[34]. Once the DPC is removed by the UPP, the DPC-occluded seDSB is exposed for the subsequent repair by homologous recombination due to lack of a second DNA terminus for end-joining[35,36]. Neutral comet assays in HCT116 cells synchronized in S-phase showed that blocking neddylation or the proteasome decreased in SN38-induced seDSBs, indicating a role of neddylation for liberating TOP1-DPC-concealed seDSBs presumably through the UPP (Fig. 3f).

The UPP is also required for the activation of DNA damage responses (DDRs) such as γH2AX once it exposes TOP1-DPC-occluded seDSBs[35]. To determine whether neddylation is involved in the activation of DDR, we carried out WB for phosphorylated CHK1 (pCHK1), single-stranded replication protein A (pRPA) and histone H2AX (γH2AX). Akin to inhibition of the proteasome, inhibition of neddylation by PEV decreased SN38-induced checkpoint response as demonstrated by measuring pCHK1 (ser345), pRPA32 (ser4/ser8) and γH2AX levels (Fig. 3g).

## CRL4 is activated by NEDD8 in response to TOP1-DPCs and targets them for ubiquitylation

As NEDD8 substrates, the cullin (CUL) family serves as scaffold proteins that provide physical support for RING ubiquitin ligases (RBX1 and 2) and substrate receptors, which form cullin-RING ligase complexes that ubiquitylate a broad spectrum of cellular proteins for proteasomal degradation[37]. Auto-mono-neddylation of cullins by their RING ubiquitin ligases (RBX1 and RBX2) results in conformation alteration and activation of the complex to facilitate the transfer of ubiquitin molecules from E2 to the substrates.

To determine whether the cullins are involved in the ubiquitylation of TOP1-DPCs, we investigated the seven cullin family members (CUL1, CUL2, CUL3, CUL4A, CUL4B, CUL5 and CUL7) that reportedly

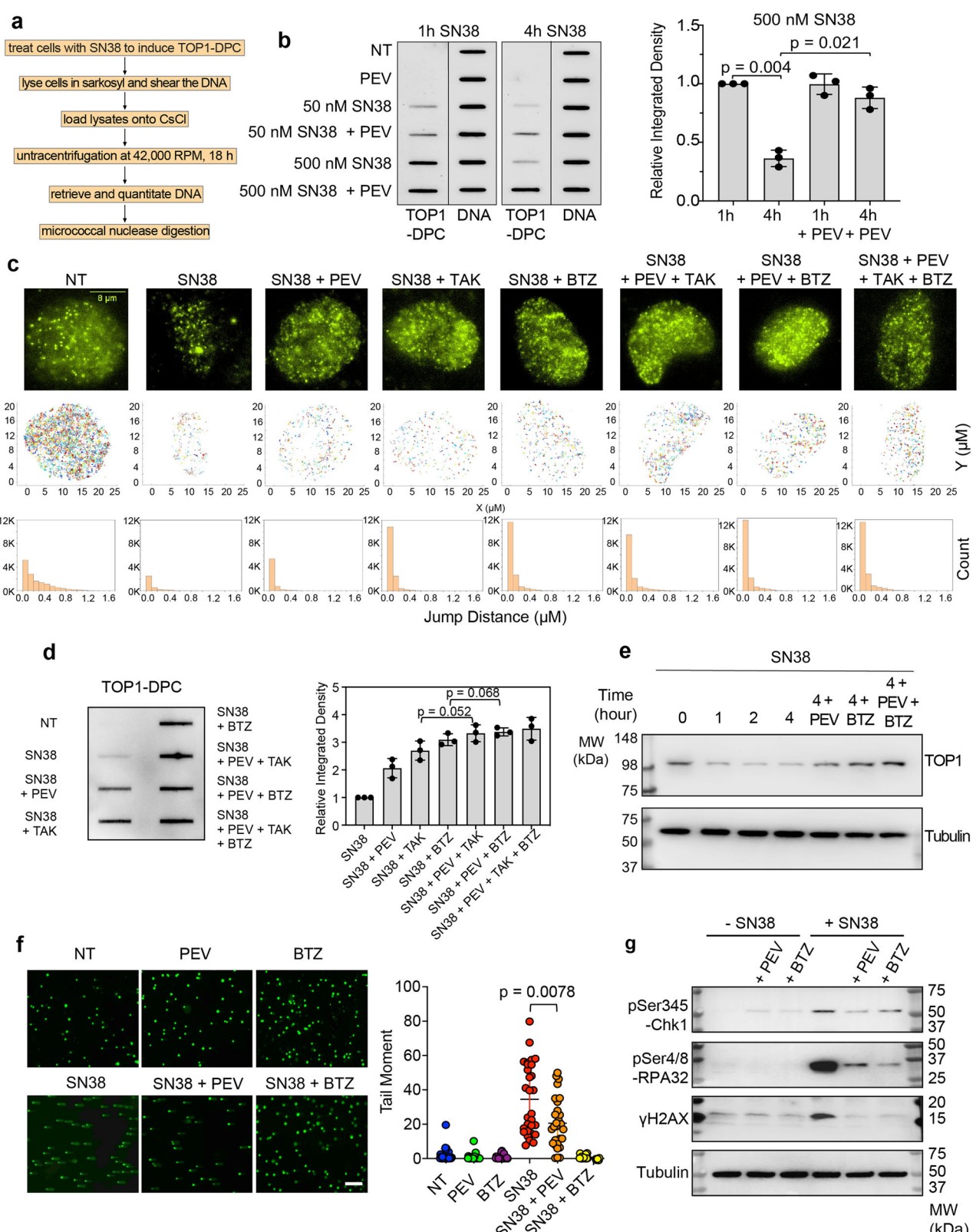

reside in the nucleus. Using the DUST (Detection of Ubiquitylated and SUMOylated TOP-DPCs) assay[14,36], we found that upregulation of CUL3, CUL4A and CUL4B in RBX1-overexpressing HCT116 cells led to increased ubiquitylation of CPT-induced TOP1-DPCs. Furthermore, CUL4A and B upregulation resulted in higher levels of ubiquitylated TOP1-DPCs than did CUL3 upregulation (Fig. 4a and Supplementary Data Fig. 4a, b).

To validate the role of CUL4 in ubiquitylating TOP1-DPCs in vitro, we performed ubiquitin assembly assays[14]. Recombinant CUL4A-RBX1 (CRL4) complex together with its E2 ubiquitin conjugating enzyme UBCH5A catalyzed polyubiquitylation of TOP1-DPCs in a dose-dependent manner (Fig. 4b). Conversely, silencing of CUL4A by siRNA or deletion of CUL4B by CRISPR in HCT116 cells led to a reduction in ubiquitylation of TOP1-DPCs (Fig. 4c, d and

**Fig. 3 | Neddylation fosters the repair of TOP1-DPCs in a manner epistatic to the ubiquitin-proteasome pathway. a** Scheme of the in vivo complex of enzyme (ICE) bioassay. **b** Left panel: ICE bioassay shows that SN38-induced TOP1 were removed after 4 h exposure to SN38 in HCT116 cells, and that pre-treatment with 10 µM PEV for 1 h blocked the removal of TOP1-DPCs. DNA loading of each sample was confirmed using anti-DNA antibody. Right panel: densitometric analyses of TOP1-DPCs from triplicate experiments including blots in the left panel. Density of TOP1-DPC/density of DNA of each group was normalized to that of cells treated with 500 nM SN38 for 1 h. Data are presented as mean ± SD, $N = 3$ biologically independent experiments. The $p$ values in this figure were calculated using two-tailed Student's $t$ test. **c** Top panels: 20 s filming of TOP1-HaloTag single molecules in HCT116 cells. The cells were divided into indicated treatments. Middle panels: plots of tracks of TOP1-HaloTag single molecules as shown in the top panels. Bottom panels: count of jumps of TOP1-HaloTag single molecules derived from the top-panel films. The bin

size is 0.1 µm. **d** Left panel: ICE bioassay in HCT116 cells treated with the indicated drug combinations for 2 h. Right panel: densitometric analyses of TOP1-DPCs. Density of TOP1-DPC/density of DNA of each group was normalized to that of cells treated with SN38 alone. Data are presented as mean ± SD, $N = 3$ biologically independent experiments. **e** HCT116 cells were treated with SN38 (10 µM) and collected at indicated time points for Western blotting. Cells were then lysed with the neutral lysis procedure. The scale bar represents 300 µm. **f** Left panel: HCT116 were treated with SN38 (10 µM) for 2 h in the presence and absence of indicated inhibitors. Cells were then subjected to neutral comet assay. Right panel: quantitation of tail moments for comet assay samples using OpenComet. Data are presented as mean ± SD, $n = 180$ total cells. Biological independent experiments were repeated three times. **g** HCT116 cells were treated with SN38 (10 µM) for 2 h. PEV (10 µM) and BTZ (1 µM) were added 1 h prior to the SN38 treatment. Cells were then subjected to Western blotting using indicated antibodies.

Supplementary Data Fig. 4c, d). These results demonstrate the role of CUL4 as an important ubiquitin ligase for TOP1-DPCs.

To examine whether CRL4-mediated ubiquitylation requires its activation by neddylation, we performed DUST assays in HCT116 cells overexpressing CRL4A (CUL4A + RBX1) or CRL4B (CUL4B + RBX1). PEV treatment suppressed CUL4A- and CULB-upregulated TOP1-DPC ubiquitylation (Fig. 4c, d), indicating that CRL4A and CRL4B ubiquitylate TOP1-DPCs in a neddylation-dependent manner. In addition, inhibition of the proteasome by BTZ in CUL4B KO cells led to an elevation of total and ubiquitylated TOP1-DPCs (Fig. 4d), suggesting the existence of additional ubiquitin ligases that ubiquitylate TOP1-DPCs and induce their degradation. Knocking-down CUL4A in CUL4B KO HCT116 cells conferred hypersensitivity to CPT (Supplementary Data Fig. 4e), substantiating the role of CUL4-dependent ubiquitylation in the repair of TOP1-induced DNA damage. We also found that the mCRC PDO #3 displayed higher levels of CUL4 proteins (Supplementary Data Fig. 4f), explaining why PDO #3 is more resistant to SN38 and PEV than the other two PDOs. We also assessed 7 cancer cell lines with different CUL4A and B expressions and observed an inverse correlation between CUL4 protein levels and SN38 sensitivity (Supplementary Data Fig. 4g, h), indicating CUL4 proteins as potential predictive biomarkers for TOP1 inhibitors.

To directly assess whether CUL4A and B are neddylated in response to TOP-DPCs, we conducted immunoprecipitation (IP) assays in HCT116 cells transfected with myc-tagged CUL4A or CULB expression plasmids with HA-tagged NEDD8 expression plasmid. Mono-neddylation of CUL4A and B was stimulated by CPT treatment within 30 min (Fig. 4e, f), demonstrating a role of CRL4 neddylation as a prompt response to TOP1-DPCs. Consistently, we also found that PEV inhibited both CUL4A and B neddylation, and that mutation in the reported CUL4A and CULB neddylation sites (K705R and K859R) blocked their neddylation[38] (Fig. 4e, f). In consonance with these findings, DUST assays showed that transfection with CUL4A K705R and CUL4B K859R constructs in HCT116 cells failed to stimulate the CPT-induced ubiquitylation of TOP1-DPCs (Fig. 4g), substantiating the essential role of CUL4 neddylation in the ubiquitylation of TOP1-DPCs.

## CRL4 ubiquitylates TOP1-DPCs for proteasomal degradation in a replication-dependent fashion

Prompted by our earlier work that revealed a replication-independent SUMO-ubiquitin pathway mediated by the SUMO ligase PIAS4 and SUMO-targeted Ub ligase (STUbL) RNF4[20,39,40], we assessed whether the CRL4-mediated ubiquitylation pathway is also activated upon SUMOylation. By performing the DUST assay in HCT116 cells pre-treated with ML-792, a potent inhibitor of SUMO-activating enzymes[41], we found that CRL4B-mediated TOP1-DPC ubiquitylation was not dependent on SUMOylation (Supplementary Data Fig. 5a, b).

Given the critical role of CRL4A and B in the regulation of DNA replication[42,43], we tested whether CRL4 ubiquitylates TOP1-DPCs in a replication-coupled manner. Consistent with this possibility, DUST

assays showed that CRL4A- and B-mediated TOP1-DPC ubiquitylation was markedly alleviated in cells pre-treated with the replication inhibitor aphidicolin (APH) (Fig. 5a). These results demonstrate that CRL4-mediated TOP1-DPC ubiquitylation is contingent on active replication.

Next, we determined the type (s) of CUL4-induced ubiquitylation linkage of TOP1-DPC. Because we previously reported that TOP1-DPCs are primarily modified by K48 and K36 polyubiquitylation[36], we transfected HCT116 cells with single lysine Ub construct HA-Ub K48 or HA-Ub K63 for DUST assays. CRL4A and B only promoted K48 polyubiquitylation of TOP1-DPCs (Fig. 5b), indicating that CUL4B catalyzes TOP1-DPC polyubiquitylation through lysines 48 of Ub, the linkage known for signaling proteasomal degradation[20, 33].

To assess whether CRL4 engages the clearance of TOP1-DPCs through ubiquitin-dependent proteasomal degradation, we performed ICE assays in HCT116 cells and found that CRL4A and B decreased levels of CPT-induced TOP1-DPCs (Fig. 5c–f). These findings demonstrate the participation of CRL4 in the removal of TOP1-DPCs. CRL4A and CULB upregulation did not impact on the levels of TOP1-DPCs in BTZ-pre-treated cells, suggesting that CRL4 removes TOP1-DPCs through the proteasome. Of note, pre-treatment with APH thwarted CRL4-potentiated TOP1-DPC removal, indicating the dependency of CRL4 activity on active replication.

## DCAF13 connects TOP1-DPCs with CRL4 for TOP1-DPC ubiquitylation

A pivotal component of the CRL4 complex is the substrate receptor DDB1- and its CUL4-associated factor (DCAF)[44], which recognizes substrate proteins and mediate their interactions with DDB1, CUL4 and RBX1 (Fig. 6a). To date, multiple DCAF proteins have been identified, but their substrates remain largely unknown. To identify the DCAF that dictates the specificity of CRL4 for the ubiquitylation of TOP1-DPCs, we transiently expressed His-tagged TOP1 in HCT116 cells and purified TOP1 protein complexes for LC-MS/MS (Fig. 6b and Supplementary Data S5). Three known DCAF proteins, DDB2, DCAF7 and DCAF13 were found enriched in the TOP1-expressing but not in the empty vector sample. To determine which DCAF is responsible for TOP1-DPC ubiquitylation by CRL4, we carried out DUST assays in CRL4B-overexpressing HCT116 cells treated with DDB2, DCAF7 or DCAF13 siRNA. Only DCAF13 silencing led to a significant reduction in CRL4B-mediated TOP1-DPC ubiquitylation (Supplementary Data Fig. 6a, b). To validate this result, we overexpressed RBX1, CRL4A or CRL4B, downregulated DCAF13, and performed DUST assays. Upregulation of CRL4A, CRL4B or RBX1 all enhanced TOP1-DPC ubiquitylation, whereas knocking-down of DCAF13 obliterated CRL4-mediated TOP1-DPC ubiquitylation (Fig. 6c), indicating that DCAF13 is required for the CRL4A/B complex to target TOP1-DPCs. Notably, the pharmacogenomic CellMiner Cross-Database (https://discover.nci.nih.gov) shows significant correlations between the expression of DCAF13 and cytotoxicity of the clinical TOP1 inhibitors topotecan and irinotecan across cancer cell lines (Supplementary Data Fig. 6c).

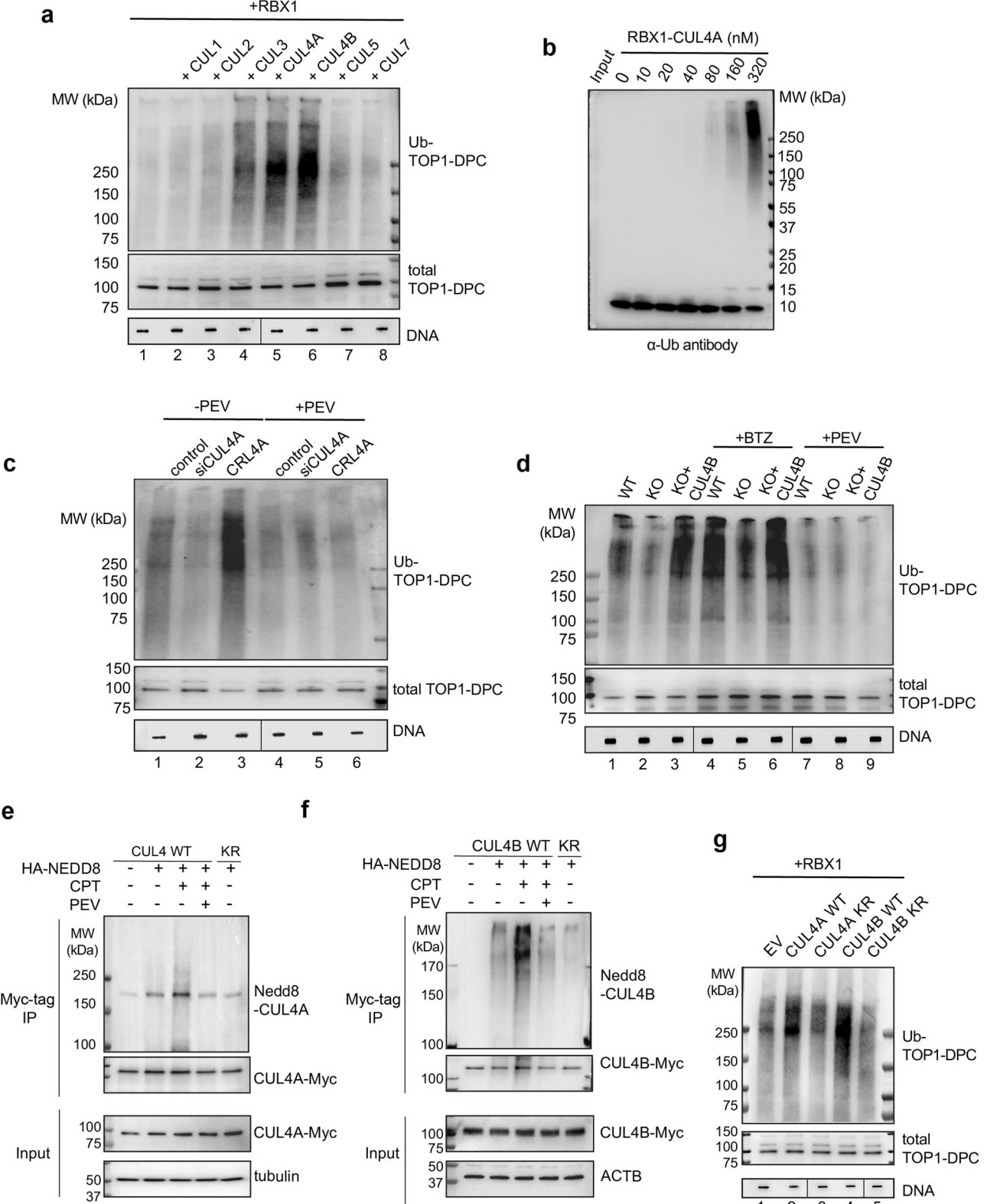

By performing immunofluorescence with a super-resolution instant structured illumination microscope (iSIM) in pre-extracted HCT116 cells, we found that DCAF13 localizes to chromatin in response to CPT, and that arresting DNA replication with APH thwarted the localization (Fig. 6d). These observations suggest that DCAF13 co-localizes with TOP1-DPCs on chromatin undergoing replication.

To determine whether TOP1-DPCs induce the binding of DCAF13 to TOP1, we pulled down His-tagged TOP1 in HCT116 cells treated with or without CPT and probed for DCAF13-FLAG. The interaction between DCAF13 and TOP1 was markedly enhanced by CPT treatment, and this interaction was diminished by APH pretreatment (Fig. 6e), These results corroborate our findings that CRL4 ubiquitylates TOP1-DPCs during DNA replication. Proximity ligation assay (PLA) in HCT116 cells overexpressing His-TOP1 and DCAF13-FLAG substantiated this replication-dependent interaction (Fig. 6f). By pulling-down DCAF13 with the endogenous DCAF13 antibody and probing the samples with a

**Fig. 4 | CRL4 is activated by NEDD8 in response to TOP1-DPCs and target TOP1-DPCs for ubiquitylation. a** HCT116 cells were transfected with the indicated Myc-tag cullin overexpression plasmids plus RBX1-FLAG overexpression plasmid for 48 h before CPT treatment (20 μM, 30 min). The cells were subjected to the DUST assay for immunodetection of ubiquitylated TOP1-DPC and total TOP1-DPC using anti-ubiquitin (Ub) and anti-TOP1 antibodies. Total DNA was detected using anti-DNA antibody as loading control. The order of DNA slot blots has been altered, as indicated by the line, and that uncropped labeled blots can be found in the Source Data file. **b** In vitro ubiquitylation assay with recombinant TOP1-DPC (generated using a suicidal DNA substrate[14]) and CUL4A-RBX1 complex. TOP1 was tested for ubiquitin conjugation in the presence of Ub E1, Ub E2 UbcH5a and the indicated concentrations of CUL4A-RBX1 complex. Reaction products were separated by SDS-PAGE and monitored by IB using anti-ubiquitin antibody. **c** HCT116 cells were transfected with CUL4A siRNA or Myc-CUL4A and RBX1-FLAG overexpression plasmids (CRL4) for 48 h before PEV pre-treatment (10 μM, 1 h) then co-treatment with CPT (20 μM, 30 min). The cells were subjected to DUST assay for immunodetection of ubiquitylated TOP1-DPC and total TOP1-DPC using anti-ubiquitin and anti-TOP1 antibodies. The order of DNA slot blots has been altered, as indicated by the line, and that uncropped labeled blots can be found in the Source Data file. **d** HCT116 cells WT, CUL4B KO cells and CUL4B KO cells replenished with Myc-

CUL4B overexpression plasmid were pre-treated with BTZ (1 μM, 4 h) or PEV (10 μM, 4 h) then co-treated with CPT (20 μM, 30 min). The cells were then subjected to DUST assay for immunodetection of ubiquitylated TOP1-DPC and total TOP1-DPC using anti-ubiquitin and anti-TOP1 antibodies. The order of DNA slot blots has been altered, as indicated by the line, and that uncropped labeled blots can be found in the Source Data file. **e** Myc-CUL4A WT or K705R (KR) overexpressing HCT116 cells were transfected with HA-NEDD8 overexpression plasmid, followed by treatments with indicated inhibitors. Immunoprecipitation (IP) using anti-Myc tag antibody was performed after the treatments. IP samples and cell lysates (input) were subjected to immunoblotting (IB) with indicated antibodies. **f** Myc-CUL4B WT or K859R (KR) overexpressing HCT116 cells were transfected with HA-NEDD8 overexpression plasmid, followed by treatments with indicated inhibitors. IP using anti-Myc tag antibody was performed after the treatments. IP samples and input were subjected to IB with indicated antibodies. **g** RBX1-FLAG overexpressing HCT116 cells were transfected with indicated CUL4 overexpression plasmids, followed by CPT treatment (20 μM, 30 min) for DUST assay for immunodetection of ubiquitylated TOP1-DPC and total TOP1-DPC using anti-ubiquitin and anti-TOP1 antibodies. The order of DNA slot blots has been altered, as indicated by the line, and that uncropped labeled blots can be found in the Source Data file.

TOP1-DPC-targeting antibody[45], we found that DCAF13 interacts with TOP1-DPCs upon SN38 treatment (Supplementary Data Fig. 6d).

iPOND analyses[46] also showed that TOP1 was immobilized on nascent DNA upon exposure to CPT (TOP1-DPCs ahead of replication forks) and co-localized with DCAF13 (Fig. 6g). Although low levels of TOP1 were trapped on mature chromatin as observed by thymidine chase, DCAF13 was not found to accompany the mature chromatin-bound TOP1 (Fig. 6g), implying that collision between replication forks and TOP1-DPCs recruits DCAF13-CRL4.

### DCAF13 interacts with the core domain of TOP1 through its putative WD40 domains

TOP1 comprises four main domains: the N-terminal domain, the core domain, the linker domain and the C-terminal domain that bears the catalytic tyrosine[8]. To determine which domain interacts with DCAF13, we expressed 6 different His-TOP1 constructs: N-terminus-truncated TOP1 (ΔN), core domain-truncated TOP1 (ΔCD), C-terminus-truncated TOP1 (ΔC), TOP1 N-terminus (N) and TOP1 core domain (CD) for His-tag pull-down assays (Fig. 7a). The core domain-truncated TOP1 and C-terminus-truncated TOP1 failed to bind DCAF13, while the TOP1 core domain alone was able to bind DCAF13, albeit not as strongly as the full-length TOP1 (Fig. 7b). This is presumably due to its lost ability to form TOP1ccs. These results suggest that the attachment of TOP1 to duplex DNA with its clamp formed by the core domain[47] is targeted by DCAF13-CRL4.

As DCAF proteins bear WD40 repeats that fold around a central axis into a propeller-like structure to contact substrate proteins[44], we modeled the WD40 domains of DCAF13 using AlphaFold2[48] and aligned the structure with its *S. cerevisiae* ortholog SOF1 (PDB: 6ZQB_46). We found 7 putative WD40 domains (Fig. 7c) conserved from yeast to human. Next, we generated DCAF13-FLAG expression constructs with deletion of each single WD40 repeat for FLAG-IP to determine the requirement of the individual WD40 motifs for TOP1 interaction. All putative WD40 repeats except WD40 repeat #1 were required for the binding of DCAF13 to TOP1 (Fig. 7d).

To determine whether the putative WD40 motifs are required for TOP1-DPC ubiquitylation, we attempted to transfect the WD40 mutants in DCAF13 KO CRC cells. We failed to generate DCAF13 CRISPR KO in CRC cells but succeeded in HEK293 cells (Supplementary Data Fig. 7a–c), suggesting the essentiality of DCAF13 for CRC proliferation. By overexpressing the WD40 mutants in HEK293 DCAF13 KO cells and performing DUST assays, we found that none of the mutants was able to induce the TOP1-DPC ubiquitylation observed in cells transfected with WT DCAF13 (Fig. 7e). These findings suggest an essential role of

the WD40 repeats of DCAF13 for the binding to and ubiquitylation of TOP-DPCs. To elucidate the role of DCAF13 WD40 repeats, we next transfected the WD40 repeat #2 Δ mutant into HCT116 cells and conducted IF using iSIM following pre-extraction. This mutant failed to localize to chromatin upon CPT treatment (Supplementary Data Fig. 7d). Together, these experiments suggest that the WD40 repeats of DCAF13 are critical for its binding to TOP1-DPCs and the subsequent ubiquitylation by CRL4 during DNA replication.

## Discussion

Our study uncovers a role of neddylation in the repair of irreversible TOP1-DPCs, one of the most frequent and deleterious DNA lesions[5]. We demonstrate that inhibition of neddylation with the clinical NAE inhibitor pevonedistat (PEV) renders CRC cells hypersensitive to the TOP1 inhibitors extensively used in CRC treatment[4, 49]. Using preclinical models including HCT116-derived mouse xenografts and metastatic CRC PDOs, we demonstrate the synergy of irinotecan/SN38 in combination with pevonedistat. Mechanistically, we found that the CUL4-RBX1 ubiquitin ligase complex (CRL4) catalyzes the K48 poly-ubiquitylation of TOP1-DPCs, which precedes their proteasomal degradation, and that CRL4 is activated upon its neddylation on lysine 705 of CUL4A and lysine 859 of CUL4B. As summarized in Fig. 7f, we propose that the CRL4 pathway is coupled with replication, as evidenced by our finding that inhibition of replication prevents CRL4-mediated ubiquitylation of TOP1-DPCs. We identify DCAF13, an understudied DDB1- and CUL4-associated factor, as the TOP1-DPC receptor in replicating DNA, allowing the adapter DDB1 in complex with CRL4 to target the DPCs for ubiquitylation, and show that DCAF13 interacts with the core domain of TOP1 via its putative WD40 repeats. A pivotal step in this CRL4-mediated UPP is the auto-mono-neddylation of CUL4, which changes the conformation of CUL4 to enable the transfer of ubiquitin moieties from RBX1 to the DPC (Fig. 7f).

This NEDD8-activated pathway appears specific for TOP1-DPCs, for we did not observed synergy of PEV with TOP2 and PARP inhibitors. TOP1 inhibitors are selectively toxic during DNA synthesis[4], which is also known to be tightly regulated by the CRL ubiquitin ligase family[42]. In addition to its implication in DNA repair, PEV elicits re-replication in part by preventing CRL4-mediated ubiquitylation and degradation of CDT1, a key licensing factor in the assembly of pre-replication complexes[32]. This raises the possibility that PEV enhances re-replication therefore collisions between replication forks and TOP1-DPCs, resulting in increased cell death as the damage cannot be repaired by the inactivated CRL4. Our results suggests that the CUL4 pathway acts in parallel with the replication-associated

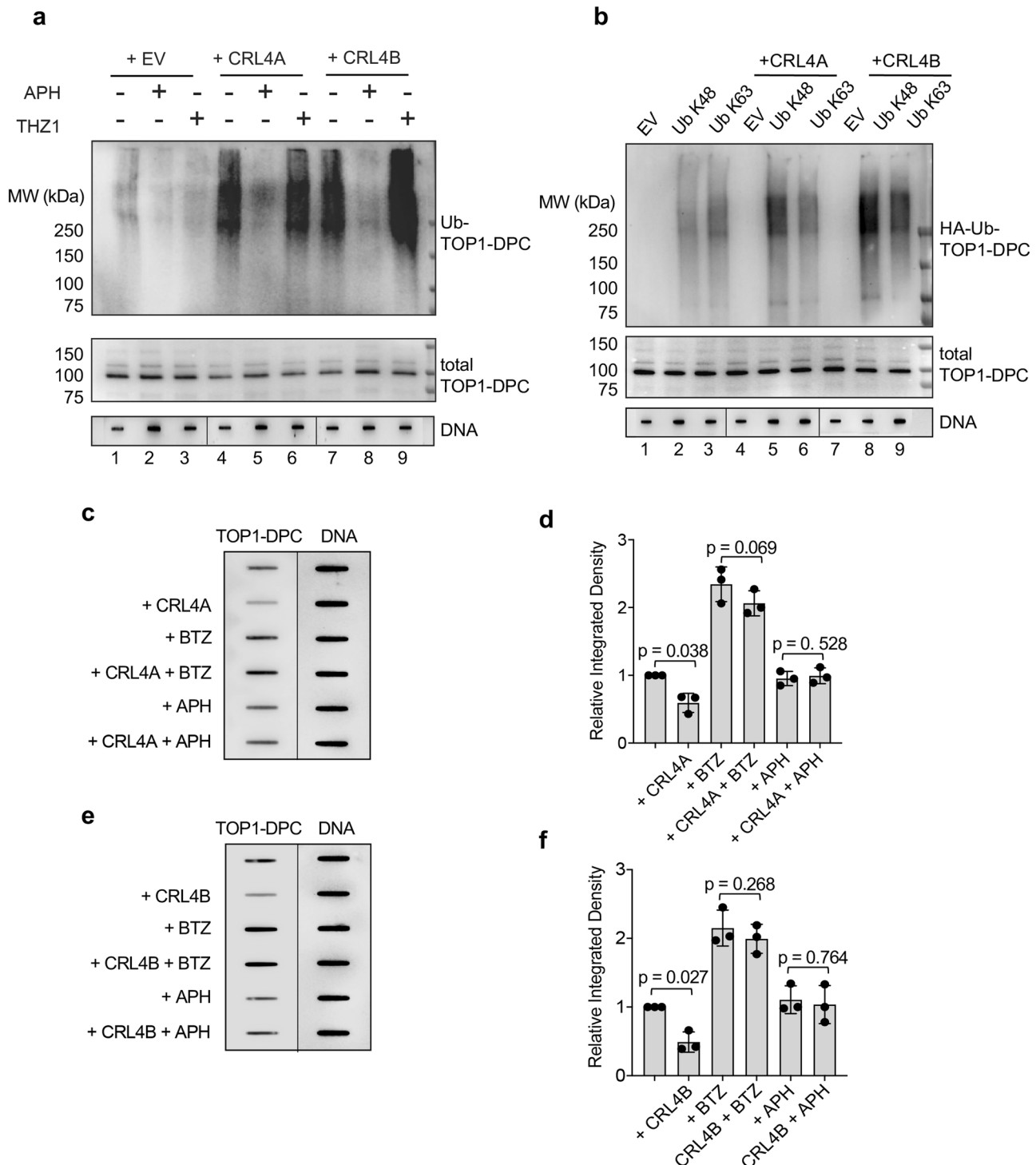

metalloprotease SPRTN, whereby SPRTN digests TOP2α-DPCs as well as TOP1-DPCs without the need for polyubiquitylation[18,19,50]. As opposed to CUL4 and DCAF13, which are highly expressed across tissues, many human tissues including colon do not express high levels of SPRTN protein (https://www.proteinatlas.org/ENSG00000010072-SPRTN/tissue). It is therefore reasonable to speculate that the proteolysis of replication-associated TOP1-DPCs is primarily driven by the UPP in certain cancers such as CRC due to their low SPRTN expression.

The identification of DCAF13 as the substrate receptor of CRL4-mediated TOP1-DPC ubiquitylation provides a new piece in the puzzle of TOP1-DPC repair. We provide evidence that DCAF13 is recruited to chromatin in response to TOP1-DPCs and localizes with nascent DNA,

suggesting that DCAF13 may possess DNA binding capacity. Implicit in these findings is that DCAF13 recognizes and binds the broken end on the newly synthesized DNA strand to target the adjacent TOP1-DPC in a way akin to DCAFs DDB2 and CSA, which sense damaged DNA to target their substrates (XPC and CSB, respectively) for CRL4-mediated ubiquitylation to promote global nucleotide-excision repair (NER) and transcription-coupled NER, respectively. It was originally reported that DCAF13 possesses RNA-binding activity and mediates rRNA metabolisms during mammalian oocyte growth[51,52]. Yet, it remains unknown whether DCAF13 directly binds DNA. The IF experiment using DCAF13 WD40 repeat #2 Δ suggests that DCAF13 localization to chromatin is likely via its tight interaction with TOP1-DPCs. Given that DCAF13 plays

**Fig. 5 | CRL4 ubiquitylates TOP1-DPCs for proteasomal degradation in a replication-dependent manner. a** HCT116 cells were transfected with empty vector (EV), Myc-CUL4A + RBX1-FLAG overexpression plasmids (CRL4A) or Myc-CUL4B + RBX1-FLAG overexpression plasmids (CRL4B) for 48 h. The cells were subjected to 1 h pre-treatment with replication inhibitor aphidicolin (APH, 10 μM) or CDK7/transcription inhibitor THZ1 (10 μM), followed by co-treatment with CPT (20 μM, 30 min). The cells were subjected to DUST assay for immunodetection of ubiquitylated TOP1-DPC and total TOP1-DPC using anti-ubiquitin and anti-TOP1 antibodies. The order of DNA slot blots has been altered, as indicated by the line, and that uncropped labeled blots can be found in the Source Data file. **b** HCT116 cells were transfected with the Ub K48 or K63 single lysine overexpression plasmid and Myc-CUL4A or B + RBX1-FLAG overexpression plasmids for 48 h before CPT treatment (20 μM, 30 min). The cells were then subjected to DUST assay for immunodetection of ubiquitylated TOP1-DPC and total TOP1-DPC using anti-ubiquitin and anti-TOP1 antibodies. The order of DNA slot blots has been altered, as indicated by the line, and that uncropped labeled blots can be found in the Source Data file. **c** HCT116 transfected with EV or Myc-CUL4A + RBX1-FLAG overexpression plasmids were pre-treated with BTZ (1 μM) or APH (10 μM) for 1 h before co-treatment with CPT (500 nM) for 2 h. The cells were then subjected to ICE assay for immunodetection of TOP1-DPC using anti-TOP1 antibody. The order of DNA slot blots has been altered, as indicated by the line, and that uncropped labeled blots can be found in the Source Data file. **d** Densitometric analyses of TOP1-DPCs from triplicate experiments including blots in panel C. Density of TOP1-DPC/density of DNA of each group was normalized to that of cells transfected with EV only. Data are presented as mean ± SD, $N = 3$ biologically independent experiments. The $p$ value was calculated using two-tailed Student's $t$ test. **e** HCT116 transfected with EV or Myc-CUL4A + RBX1-FLAG overexpression plasmids were pre-treated with BTZ (1 μM) or APH (10 μM) for 1 h before co-treatment with CPT (500 nM) for 2 h. The cells were then subjected to ICE assay for immunodetection of TOP1-DPC using anti-TOP1 antibody. **f** Densitometric analyses of TOP1-DPCs from triplicate experiments including blots in (**e**). Density of TOP1-DPC/density of DNA of each group was normalized to that of cells transfected with EV only. Data are presented as mean ± SD, $N = 3$ biologically independent experiments. The $p$ value was calculated using two-tailed Student's $t$ test.

a crucial role in the regulation of rDNA processing in the nucleolus, whether the DCAF13-DDB1-CUL4 complex is recruited upon collision between the RNA polymerase I complex and the TOP1-DPC therefore warrants future investigations[53].

In addition to our findings showing a direct role of neddylation for TOP1-DPC repair, neddylation has been proposed to regulate DDRs. Neddylation has been shown to modify CRLs for DNA damage checkpoint control (SKP1-CUL1-F-box), DDRs (DDB1-CUL4-CDT2)[54] and nucleotide excision repair control (DDB1-CUL4-DDB2 and -CSA)[55]. Neddylation has also been reported to facilitate CRL-mediated Ku70/80 ubiquitylation to ensure DSB repair by NHEJ[56]. Another known target of neddylation involved in DDRs is histone H4, whose DSB-induced polyneddylation appear to play a crucial role in recruiting RNF168 and its functional partners RNF8, 53BP1 and BRCA1 to the damaged site for homologous recombination[25]. Whether TOP1-DPCs trigger histone H4 neddylation requires further exploration.

Targeting the proteasome as a strategy to enhance the activity of TOP1 inhibitors has been proposed. Yet, early phase II studies failed to show improved activity of irinotecan in combination with proteasome inhibitors in patients with relapsed or refractory CRC. This might be ascribable to the broad spectrum of cellular targets of the proteasome[57,58]. Our studies in CRC preclinical models suggest that selective targeting of the degradation system by neddylation inhibition could be a more effective and safer avenue to overcome TOP1 inhibitor resistance in patients. As TOP1-targetd therapies are extensively used for a variety of cancers and second-generation TOP1 inhibitors are being developed[4], insights from our study may be relevant to other cancers treated with TOP1 inhibitors such as ovarian and small cell lung cancers.

## Methods
Our research complies with all relevant ethical regulations. The Institutional Animal Care and Use Committee at NIH has approved our animal study protocol. The Institutional Review Board at NIH has approved our human sample study protocol.

### Human cell culture
Cell lines used in the study were obtained from the NCI Development Therapeutics Program. HCT116 colorectal cancer cells, HT29 colorectal cancer cells and HEK293 human embryo kidney cells were cultured in DMEM medium (Life Technologies) supplemented with 10% (v/v) fetal bovine serum, 100 units/ml penicillin, 100 μg streptomycin/ml streptomycin and 1x GlutaMax. HCT15, KM12, SW837, SW48 and SW620 colorectal cancer cells were grown in RPMI1640 + 10% (v/v) fetal bovine serum, 100 units/ml penicillin, 100 μg streptomycin/ml streptomycin and 1x GlutaMax. All the cell lines were cultured in tissue culture flasks or dishes at 37 °C in a humidified $CO_2$−regulated (5%)

incubator. All experiments were performed within 25 passages from thawing, and cell lines were routinely tested for mycoplasma contamination.

### Quantitively high-throughput and matrix screening
The MIPE 5.0 library is a collection of 2480 mechanistically annotated approved and investigational drugs (2480 small-molecules, including multiple inhibitors for well-explored oncogenic targets while simultaneously encompassing mechanistic diversity, targeting over 800 distinct mechanisms-of-action.

For high-throughput drug screening, HCT116 cells were grown in DMEM medium (Gibco) supplemented with 5% FBS, 2 mM Glutamine 100 U/ml Pen/Strep. Cells were maintained in a humidified $CO_2$ incubator at a density of 0.25/0.50 MLNs/ml before seeding. In total, 500 cells/well were then seeded into 1536 well white polystyrene tissue culture-treated plates (Greiner), in a final volume 5 μl of growth media containing either DMSO or pevonedistat (MLN-4924, TargetMol, Cat# T6332) at a predetermined set of doses (2000, 500, 125, 31.25 and 7.81 nM), by using a Multidrop Combi dispenser (Thermo Fisher). After cell addition, 23 nl of MIPE 5.0 compounds were transferred to individual wells (5 doses tested for each compound in separate wells) via a 1536 pin-tool. Bortezomib (final concentration 20.3 μM) was used as a positive control for cell cytotoxicity. Plates were covered by a stainless-steel gasketed lid to prevent evaporation and incubated for 48 h in a humidified CO2 incubator. At the 48-h time point, 3 μl of CellTiter Glo (Promega) were added to each well and plates were incubated at room temperature for 15 min with the stainless-steel lid in place. Luminescence readings were taken using a Viewlux reader (PerkinElmer) with a 2 s exposure time per plate. Viability of compound treated wells was normalized to DMSO and empty well controls present on each plate, and dose-response curve-fitting and curve-classification was automatically performed for each individual drug.

### Luciferase expressing HCT116 cell line-derived mouse xenografts
Female athymic nude mice were obtained from the NCI-Frederick Mouse Repository (MD, USA). All procedures were performed in compliance with protocols approved by the NIH Institutional Animal Care and Use Committee and were in accordance with federal guidelines for the humane treatment and care of laboratory animals. Mice were randomly assigned into control or treatment groups, without blinding. Sex was not considered in the study design and analysis.

HCT116 cells (0.75 Mio cells/mouse) were subcutaneously transplanted into the left flank of 6−8-week-old female athymic nude mice in a 1:1 mix of PBS and Matrigel. Tumor size and mouse weight were assessed weekly. Tumor volume was calculated by the formula:

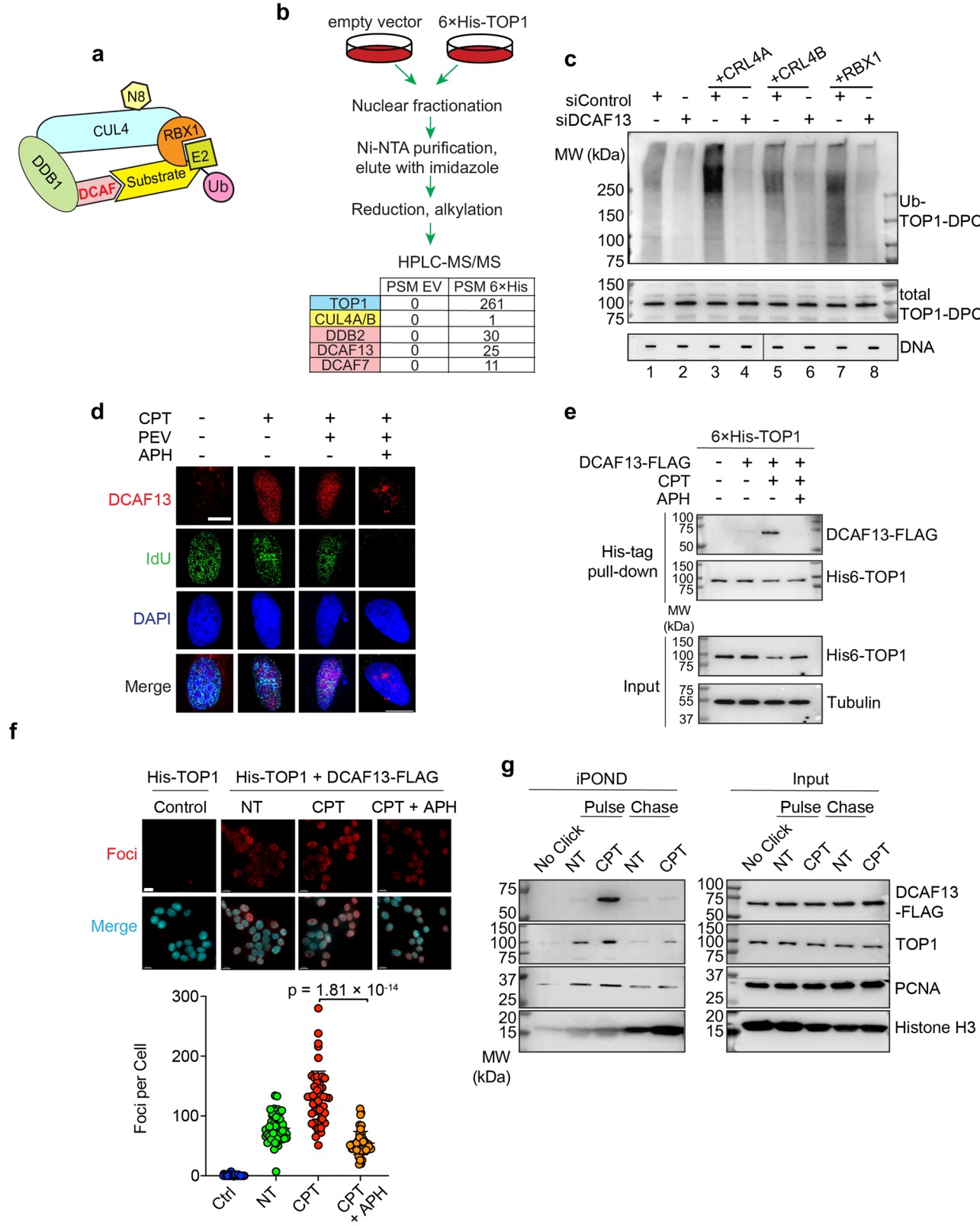

volume = 0.5 × length × width[2]. When tumor size reached 95.7 mm[3], treatment was initiated as shown in Supplementary Data Fig. 2b. The maximal tumor size must not exceed 20 mm at the largest diameter, and the maximal tumor size in this study was not exceeded.

Mice were injected 2 days after final treatment (d21) with D-luciferin (150 mg/kg body weight) intraperitoneally. Images were taken 10 min after injection using a Xenogen IVIS System. Signal intensity quantification was performed using the Living Image software (Xenogen).

**Toxicity study in mice**

The study protocol was approved by the NCI Animal Care and Use Committee. Female athymic nude mice were obtained from Charles River. Four treatment groups (n = 5: 2 mice for mid-point

**Fig. 6 | DCAF13 connects TOP1-DPCs with CRL4 for ubiquitylation. a** A model for the assembly of the RBX1-CUL4-DDB1-DCAF ubiquitin complex. N8 NEDD8, U ubiquitin. **b** Scheme of 6×His-TOP1 pull-down in HCT116 cells for LC-MS/MS. Three DCAF member proteins were enriched in the 6×His-TOP1 overexpressing sample. PSM peptide spectrum match. **c** HCT116 cells were transfected with indicated plasmids and control siRNA (siControl) or DCAF13 siRNA, followed by CPT treatment (20 µM, 30 min) for DUST assay immunodetection of ubiquitylated TOP1-DPC and total TOP1-DPC using anti-ubiquitin and anti-TOP1 antibodies. **d** Immunofluorescence in IdU-labeled DCAF13-FLAG overexpressing HCT116 cells. Cells were pre-extraction and DCAF13 foci and IdU foci were monitored by instant structured illumination microscope (iSIM) using anti-FLAG and anti-BrdU antibodies. The scale bar represents 10 µm. Biological independent experiments were repeated two times. **e** 6×His-tagged TOP1-expressing HCT116 cells were transfected with DCAF13-FLAG overexpression plasmid or empty vector, followed by treatments with DMSO or CPT (20 µM) ± APH (10 µM, pre-treatment for 2 h) for 30 min. His-pull-down samples and cell lysates (input) were subjected to immunoblotting (IB) with indicated antibodies. **f** Top panel: HCT116 cells were transfected with the indicated plasmids before DMSO or CPT treatment (20 µM, 30 min), followed by proximity ligation assay (PLA) using anti-His antibody and anti-FLAG antibody. The scale bar represents 15 µm. Bottom panel: quantitation of foci per cells of each treatment group as shown in the upper panel. Data are presented as mean ± SD, $n = 222$ total cells. The $p$ value was calculated using two-tailed Student's $t$ test. Experiments were repeated three times. **g** DCAF13-FLAG expressing HCT116 cells were pulse-labeled with EdU for 10 min in the absence or presence of CPT (1 µM), then chased with thymidine for 40 min, followed by iPOND analysis. iPOND pull-down samples and cellular lysates (IP) were subjected to IB with indicated antibodies. In no-click samples, desthiobiotin-TEG azide was replaced by DMSO.

assessments, 3 mice for end of study assessments): (1) Vehicle control, IP daily M-F for 3 weeks; (2) Irinotecan 20 mg/kg IP qwk × 3 weeks; (3) Pevonedistat 2.5 mg/kg, IP daily M-F for 3 weeks; (4) Irinotecan 20 mg/kg IP qwk × 3 weeks/Pevonedistat 2.5 mg/kg, IP daily M-F for 3 weeks.

Both drugs were dissolved in the same vehicle: 5% DMSO, 20% PEG200, 5% Tween80 and will be administered at 0.1 ml/10 g BW. Mice received intraperitoneal treatment of either vehicle or the respective drugs for 3 cycles.

Mice were monitored and weighed daily (M-F). At mid-point in the study (~day 10), 2 animals per group were sacrificed to collect whole blood via cardiac puncture and organ weights (spleen, liver, lung, heart, kidneys, brain). At end of study (~day 22), the remaining 3 mice per group were sacrificed to collect whole blood via cardiac puncture and organ weights (spleen, liver, lung, heart, kidneys, brain). Blood samples were immediately processed on day of collection to analyze for CBC/chemistry. Absolute and relative organs weights, automated hematology (CBC), and clinical chemistry (heparinized plasma) was recorded for each mouse.

### Patient-derived organoid (PDO) culture and maintenance

In brief, PDOs #1 and #2 were generated from surgical resection of colorectal metastases following patient consent, NIH institutional review board (IRB) and ethical approval. The fresh tumor was mechanically and enzymatically digested using a tumor disassociation kit (Miltenyi Biotec). Single cells were suspended in 80% Matrigel and 20% colorectal cancer-specific media, which was adapted from prior publications[59], and plated in tissue culture-treated plates. Warmed colorectal cancer-specific media was added to the plates after solidification of the Matrigel and incubated in 37 °C. These PDO lines were passaged every 2–3 weeks as necessary as per previously published protocols[60]. CRC PDO #3 was generated from a human CRC liver metastasis biopsy specimen following patient consent, NIH institutional review board (IRB) and ethical approval. The pathological specimens were immediately stored in storage media (1× DMEM/F12, 1× Glutamax and 10 mM HEPS buffer) on ice. The tissues were immediately subjected to enzymatic disassociation. Briefly, PDOs were culture in drop of growth factor reduced Basement Membrane Extract (BME) (Corning) and medium was refreshed every 4 days. The culture media contains, DMEM/F12 (Gibco) with 100 U/ml Penicillin/Streptomycin (Gibco), 10 mM HEPS buffer(Invitrogen), 1× Glutamax (Gibco), (100 µg/ml Primocin (Sigma), 1 mM NAC (sigma), 50% WNT3a conditioned media, 10% RSPO1 conditioned media, 10% Noggin conditioned media, 50 ng/ml EGF (StemCell Technologies), 10 nM Gastin (Sigma), 100 nM IGF-1 (StemCell Technologies), 100 nM FGF-2 (StemCell Technologies) 0.5 µg/ml A83-01 (Sigma), 1× B27 (Thermo Fisher Scientific) and 10 µM Y-27632 (StemCell Technologies). The organoids were passage through shear stress with 1 U/ml Dispase/DMEM/F12 solution (StemCell Technologies) followed by trypsin-EDTA treatment (Invitrogen)[59].

### RNA-seq

Total RNA (100 ng) was prepared for Illumina RNA sequencing. Poly(A) RNA was purified using NEBNext Poly(A) Magnetic Module (New England Biolabs, #E7490) followed by library preparation using NEBNext Ultra II Directional RNA Library Prep Kit (#E7760S), according to the manufacturer's instructions. Libraries were quantitated by qPCR, pooled, and sequenced on an Illumina NextSeq 500 System running a 150 Cycle Mid Output Kit v2.

### Drug treatment in PDOs

The CRC organoid cultures were passed 48 h prior to the drug treatment. Briefly, for drug testing, the CRC-PDOs were enzymatically disassociated using with Dispase/DMEM12 [1 U/ml] solution (StemCell technologies) followed by mechanical disassociation for 5 min at 4 °C. The disassociated PDOs were washed with excess Basal Media (DMEM/F12/1XGlutmax/1 mM HEPES), spun down the organoids by 500 g for 5 min at 4 °C. The cells in CRC organoid growth media were resuspended and kept in ice for further process. $4 \times 10^5$ cells were suspended in 1 ml cold MBM media and mixed 1 ml of ECM (Matrigel Growth Factor Reduced (Corning) in a sterile cell dispenser boat kept on ice. Dispensed 10 µl of organoids/ECM complex in prewarmed 384 well opaque plates using an Integra voyager multichannel pipette. The seeding density of PDOs was 2000 cells/well. Once the cells were seeded, the plates were spun down at 100 g for 1 min and then kept in $CO_2$ incubator for 30 min. After 30 min, the ECM/CRC media-organoids complex were solidified, and 20 µl CRC media was added on top of the organoids. The plates were gently spun down at 100 g for 1 min and then incubated at 37 °C for 46 h for the growth and development of organoids in the ECM-MBM complex. On the day of drug treatment, CRC organoid media and drugs were set at room temperature prior to the drug testing. Added 30 µl of media with 2× concentration of drug in triplicates and with corresponding vehicle controls. The plates were spun down at 100 g for 1 min and incubated at 37 °C for 72 h for the growth of organoids. Twenty µl of CellTiter-Glow luminescent cell viability reagent (Promega) were added into reach wells and incubate at room temperature on shaker for 30 min. Luminescence intensity was measure using SpectroMaxi3, CellTiter-Glo cell proliferation program and the combination index was calculated using Combenefit software[61].

### Single-molecule fluorescence microscopy

Single-molecule imaging experiments were conducted on a custom-built Nikon Ti microscope. The microscope is equipped with a ×100 Oil-immersion objective lens (N.A. = 1.49), a multi-band dichroic (405/488/561/633 BrightLine quad-band bandpass filter, Semrock, USA) and a piezo z-stage (ASI, USA), a filter wheel (Sutter Instrument, USA) and a stage top incubator (Tokai Hit, Japan). The lasers were focused into the back pupil plane of the objective to generate wide-field illumination. A Nikon N-STORM module was used steer the incidence angle of the laser for generating inclined illumination. The emission was collected

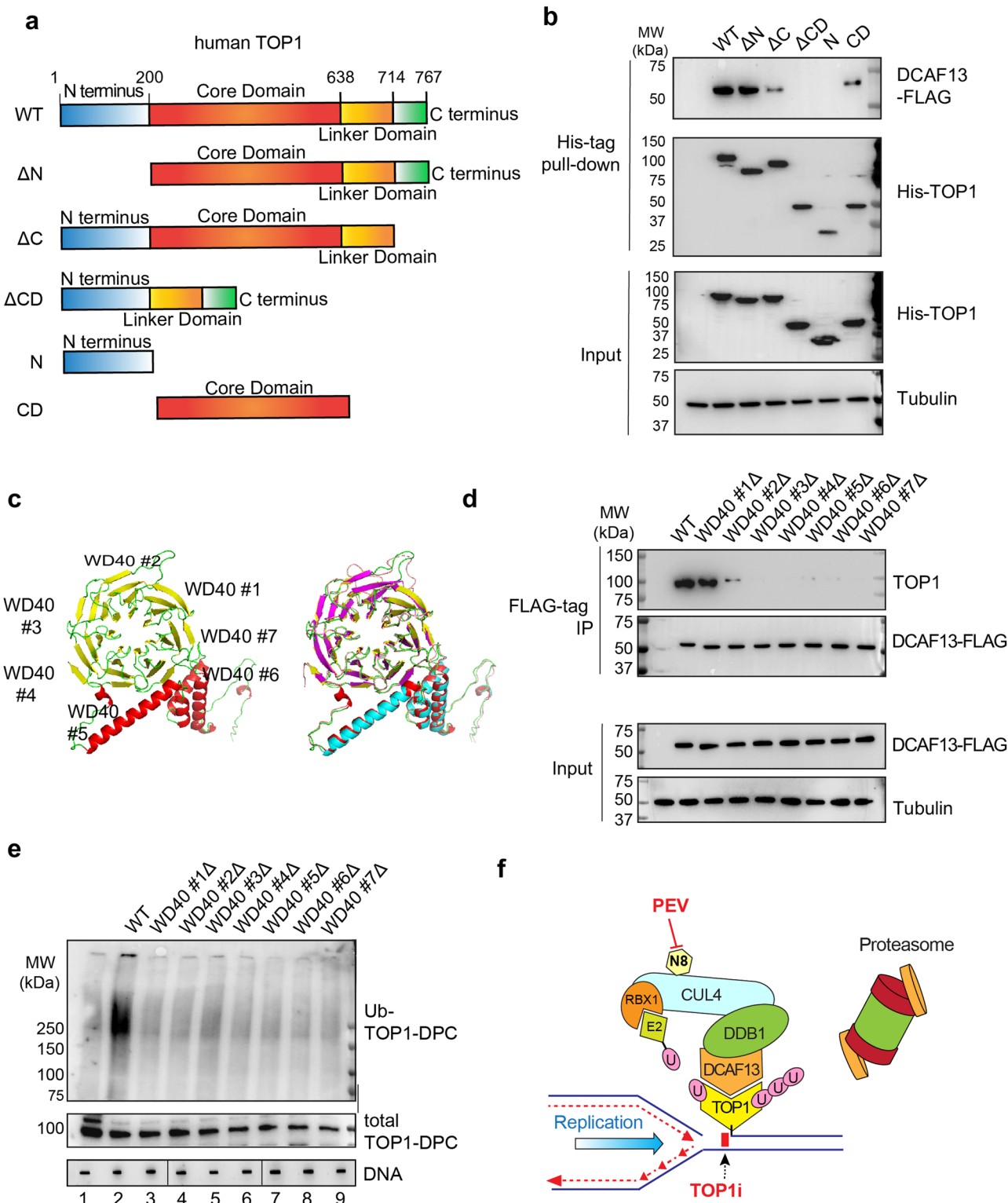

by the same objective passing through an emission filter (617/73, Semrock) in front of sCMOS camera (Prime 95B, Teledyne Photometrics). The microscope, lasers and the camera were controlled through NIS-Elements (Nikon, USA).

**Single-molecule tracking and analysis**

Single-molecule tracking was performed with custom written MATLAB software (http://site.physics.georgetown.edu/matlab/) based on available tracking algorithms. The MATLAB scripts adapted from IDL Particle Tracking were used to localize and track single molecules. The

positions of the diffraction-limited spots in the trajectories were determined with 2D Gaussian fit.

For jump distance analysis, the probability that a particle located at position $r$ at time $t$ in two dimension will be found at position $r'$ at time $t + \tau$ is given by[62]:

$$\varphi(r,t) = \left(\frac{1}{4\pi Dt}\right)\exp(-r^2/4Dt)$$

where $D$ is the diffusion constant.

**Fig. 7 | DCAF13 interacts with the core domain of TOP1 using its putative WD40 domains. a** Domain schematics of human TOP1. **b** HCT116 cells were co-transfected with DCAF13-FLAG overexpression plasmid and indicated 6×His TOP1 constructs, followed by CPT (20 µM, 30 min) in the presence of BTZ (1 µM) for His-tag pull-down. The pull-down samples and cellular lysates (input) were subjected to IB using indicated antibodies. **c** Model structure of human DCAF13 predicted by AlphaFold2[48] (yellow, green, red) superimposed with SOF1 (yeast ortholog of human DCAF13 PDB: 6ZQB_46, red, magenta, blue). **d** HCT116 cells were transfected with the indicated DCAF13-FLAG constructs, followed by CPT (20 µM, 30 min) in presence of BTZ (1 µM) for FLAG-IP. The IP samples and cellular lysates (input) were subjected to IB using indicated antibodies. **e** HCT116 cells were transfected with the indicated DCAF13-FLAG constructs followed by CPT (20 µM, 30 min) for DUST assay for immunodetection of ubiquitylated TOP1-DPC and total TOP1-DPCs using anti-ubiquitin and anti-TOP1 antibodies. The order of DNA slot blots has been altered, as indicated by the line, and that uncropped labeled blots can be found in the Source Data file. $N = 1$. **f** TOP1-DPC arrest the replication forks upon their collision and signal the DCAF13-DDB1-CUL4-RBX1 complex for its recruitment to the DPCs. DCAF13 the substrate receptor that binds TOP1-DPCs and links the DPCs to the CRL4 complexes. CUL4 is activated by mono-neddylation by RBX1, which facilitates the transfer of ubiquitin from E2 and RBX1 to TOP1-DPC. The ubiquitylation leads to proteasomal degradation of TOP1-DPCs at arrested replication forks, leading to exposure of the otherwise concealed seDSB. N8 NEDD8, U ubiquitin.

In the case of 2D diffusion, the displacement probability is obtained through integrating the above equation over the circular shell of width ($dr$):

$$p(r,t)dr = dr \int_0^{2\pi} r\varphi(r,t)d\theta = \frac{2\pi r dr}{4\pi Dt}\exp(-\frac{r^2}{4Dt})$$

Experimentally, this probability distribution can be approximated by counting the jump distances within respective intervals ($r, r+dr$) traveled by a single-molecule during a given time.

Mean square displacements (MSDs) were calculated from xy positions as previously described[63]. The tracks were computed and plotted with @msdanalyzer script[64].

## Generation of gene knockout cells using CRISPR-Cas9

To stably knockout the genes encoding CUL4B and DCAF13, the CRISPR-Cas9 genome-editing method was used[65]. To delete CUL4B in HCT116 cells, two 25-bp guide RNA sequences 5′-CACCGTAAA-CAGTAACTCTAACCTC-3′ and 5′-AAACTAGGCACATATGTCGGCCAAC-3′ targeting CUL4B exon 6 were designed using the CHOP CHOP tool and cloned into the Cas9 expressing guide RNA vectors pX458 and pX459, respectively. In brief, the Bbs1 cutting site containing guide RNA sequences were annealed and then cloned into the guide RNA vector using T4 ligase (New England Biolabs). The plasmids were transfected with Lipofectamine 3000 in HCT116 cells. Transfected cells were enriched by selection in 1 µg/ml puromycin containing media for 3 days prior to isolation of single clones and screening for loss of CUL4B by Western blotting.

To delete DCAF13 in HEK293 cells, two guide RNA sequences 5′-CACCGCGGAAGAGCAACCGAGATGA-3′ and 5′-AAACGACAATTATGTC CGCGAAACC-3′ targeting DCAF13 exon 1 were designed and cloned into pX458 and pX459, respectively. The plasmids were transfected in HEK293 cells, followed by selection in media containing 1 µg/ml pur-omycin for 3 days prior to isolation of single clones and screening for loss of DCAF13 by Western blotting.

## Expression plasmids and siRNAs

For human TOP1 expression in HCT116 cells, N-terminally 6×His-tag-ged TOP1 cDNA was amplified by PCR using primers 5′-GCATATC CTCGAGCGCCACCATGGCCCATCATCACCATCACCACAGTGGGGACC ACCTCCACAAC -3′ and 5′-GCTAGAGCGGCCGCCTAAAACTCATAGTCT TCATCAGC -3′ using pGALhTOP1 yeast plasmid as template. The PCR product was inserted into PspXI/NotI sites of a pT-REx-DEST Gateway vector (Invitrogen). HA-ubiquitin WT, K48 and K63 plasmids were a gift from Ted Dawson. Myc-CUL1, Myc-CUL2, Myc-CUL3, Myc-CUL4A, Myc-CUL4B, Myc-CUL5 and Myc-CUL7 WT plasmids were a gift from Yue Xiong. DCAF13-FLAG was purchased from OriGene (RC211029). RBX1-FLAG was purchased from OriGene (RC200348).

siRNA transfections were performed using Lipofectamine RNAi-MAX (Invitrogen) according to the manufacturer's instructions. All siRNAs were used at a final concentration of 50 nM unless otherwise indicated. The following siRNAs were used: control siRNA (Dharma-con, Cat# D-001206-13-05); CUL4A siRNA (Dharmacon, Cat# M-012610-01-0005), CUL4B siRNA (Dharmacon, Cat# M-017965-01-0005), DDB2 siRNA (Dharmacon, Cat# M-011022-01-0005), DCAF7 siRNA (Dharmacon, Cat# M-019999-01-0005), DCAF13 siRNA (Dharmacon, Cat# M-017898-01-0005), p53 siRNA (Santa Cruz Biotechnology, sc-29435).

## Site-directed mutagenesis (SDM) in mammalian expression vectors

Myc-CUL4A K705R was generated by Q5 SDM Kit (NEB) using oligo-nucleotide 5′ AGAATAATGAGGATGAGAAAGACTC. Myc-CUL4B K859R was generated by Q5 SDM Kit (NEB) using oligonucleotide 5′-CGSAATTATGAGGATGAGAAAGAC-3′. DCAF13-FLAG putative WD40 repeats #1, 2, 3, 4, 5, 6, 7Δ were generated by Q5 SDM Kit using oligonucleotides 5′-TGTATCCGTACAATACAAGC-3′, 5′-GAAGAGCCATTA CATACAATATTAG-3′, 5′-ATGACCTGGGGATTTGAC-3′, 5′-GTTATCTTA GATATGAGAACAAATAC-3′, 5′-GTCCATATGGATCATGTATC-3′, 5′-TATCATACAAAGAGAATGCAAC-3′, 5′-CTTACATCACGAGAAAAAG-3′, respectively. 6×His-tagged N-terminally truncated TOP1 (ΔN), core domain-truncated TOP1 (ΔCD), C-terminally truncated TOP1 (ΔC), TOP1 N-terminus (N) and TOP1 core domain (CD) were generated by Q5 SDM Kit using oligonucleotides 5′-AAGTGGAAATGGTGGGAAG-3′, 5′-TAGGCGGCCGCTCTAGAG-3′, 5′-ACTTTTGAGAAGTCTATGATG-3′, 5′-TAGGCGGCCGCTCTAGAG-3′ and 5′-TAGGCGGCCGCTCTAGAG-3′, respectively.

## Antibodies

Anti-α Tubulin, rat polyclonal, Santa Cruz, Cat# sc-53030; Anti-ubi-quitin, mouse monoclonal, Santa Cruz, Cat# sc-8017; Anti-phospho-Histone H2A.X (Ser139), mouse monoclonal, Millipore, Cat# 05-636-I; Anti-pRPA32 (Ser4/Ser8), rabbit polyclonal, Bethyl Lab, Cat# A300-245A; Anti-pCHK1 (Ser345), rabbit monoclonal, Cell Signaling, Cat# 2348; Anti-TOP1, mouse monoclonal, BD Biosciences, Cat# 556597; Anti-TOP1-DPC, mouse monoclonal, Millipore, Cat# MABE1084; Anti-His-tag, rabbit monoclonal, Cell Signaling, Cat# 12698; Anti-FLAG, mouse monoclonal, Sigma, Cat# F1804; Anti-FLAG, rabbit polyclonal, Sigma, Cat# F7425; Anti-Myc, mouse monoclonal, Cell Signaling, Cat# 2276; Anti-dsDNA, mouse monoclonal, Abcam, Cat# ab27156; Anti-CUL4A, rabbit polyclonal, Bethyl Lab, Cat# A300-739A; Anti-CUL4B, Novus Biological, rabbit polyclonal, Cat# H00008450-B01P; Anti-DDB2, rabbit polyclonal, Thermo Fisher, Cat# PA5-37361; Anti-DCAF7, rabbit polyclonal, Thermo Fisher, Cat# PA5-93222; Anti-DCAF13, rabbit monoclonal, Abcam, Cat# ab195121; Anti-p53, rabbit monoclonal, Cell Signaling Technology, Cat# 9282. Anti-cleaved caspase-3, rabbit polyclonal, Abcam, ab2302. Goat anti-mouse IgG (H + L) Cross-Adsorbed Secondary Antibody, Thermo Fisher, A-11001. Alexa Fluor™ 488 Goat anti-rabbit IgG (H + L) Cross-Adsorbed Secondary Antibody, Alexa Fluor™ 568, Thermo Fisher, A-11011. All the antibodies were used at 1:1000 dilution.

## Cell viability assay

To measure drug sensitivity, cultured cancer cells were continuously exposed to various concentrations of the drugs. Ten thousand cells were seeded in 96-well white plates (PerkinElmer Life Sciences,

6007680) in 100 µl of medium per well. Cells were incubated for 72 h in triplicate. Cellular viability was determined using the ATPlite 1-step kits (PerkinElmer). Briefly, 50 µl ATPlite solution was added in 96-well plates per well, respectively. After 5 min, luminescence was measured with an EnVision 2104 Multilabel Reader (PerkinElmer). The ATP level in untreated cells was defined as 100%. Viability (%) of treated cells was defined as ATP treated cells/ATP untreated cells × 100.

## Cleaved caspase-3 measurement
RayBio CASP-3 (D175) ELISA Kit was used to measure cleaved CASP-3 (Asp-175) in HT29 cell lysates following manufacturer's instructions.

## Western blotting
SN38-induced TOP1 degradation was monitored by Western blotting of the alkaline lysates prepared from drug-treated HCT116 cells with slight modifications[66]. Following treatment, cells were washed with DMEM and incubated at 37 °C in a $CO_2$ incubator for 30 min then lysed with 100 µl of an alkaline lysis buffer (200 mM NaOH, 2 mM EDTA). Alkaline lysates were neutralized by the addition of 100 µl of 1 M HEPES buffer, pH 7.3, followed by mixing with 10 µl 100 mM $CaCl_2$, 1 µl 2 M DTT and 2 µl 100× protease inhibitor cocktail and 200 units of micrococcal nuclease. The resulting mixtures were incubated on ice for 1 h. Seventy µl of 4× Laemmli buffer was added to each sample. The lysates were boiled for 10 min, analyzed by SDS-PAGE, and immunoblotted with various antibodies as indicated. Other proteins were detected by lysing cells with RIPA buffer (150 mM NaCl, 1% NP-40, 0.5% Sodium deoxycholate, 0.1% SDS, 50 mM Tris pH 7.5, 1 mM DTT and protease inhibitor cocktail). Images were acquired using ChemiDoc imager and Image Lab software (Bio-Rad).

## In vivo of complex (ICE) assay
TOP1-DPCs were isolated and detected using in vivo complex of enzyme (ICE) assay[31]. Briefly, HCT116 cells were lysed in sarkosyl solution (1% w/v) after treatment. Cell lysates were sheared through a 25 g 5/8 needle (10 strokes) to reduce the viscosity of DNA and layered onto CsCl solution (150% w/v), followed by centrifugation in NVT 65.2 rotor (Beckman coulter) at $270,000 \times g$ for 20 h at 25 °C. The resulting pellet containing nucleic acids and TOP1-DPCs was obtained and dissolved in TE buffer. The samples were quantitated and subjected to slot-blot for immunoblotting with various antibodies as indicated. Two µg of DNA is applied per sample. For mass spectrometric analysis, ICE samples were treated with RNase A to eliminate RNA contamination. Experiments were performed in triplicate and TOP1-DPCs were quantified by densitometric analysis using ImageJ.

## DUST assay
After TOP1 inhibitor treatments, $1 \times 10^6$ HCT116 cells in 35 mm dish per sample were washed with PBS and lysed with 600 µl DNAzol (Invitrogen), followed by precipitation with 300 µl 200 proof ethanol. The nucleic acids were collected, washed with 75% ethanol, resuspended in 200 µl TE buffer then heated at 65 °C for 15 min, followed by shearing with sonication (40% output for 10 s pulse and 10 s rest for 4 times). Samples were centrifuged at $20,000 \times g$ for 5 min and the supernatants were collected and treated RNase A (100 µg/ml) for 1 h, followed by addition of 1/10 volume of 3 M sodium acetate sodium acetate and 2.5 volume of 200 proof ethanol. After 20 min full speed centrifugation, DNA pellets were retrieved and resuspended in 100 µl TE buffer for spectrophotometric measurement to quantitate DNA content. Ten µg of each sample was digested with 50 units micrococcal nuclease (Thermo Fisher Scientific, 100 units/µl) in presence of 5 mM $CaCl_2$, followed by SDS-PAGE electrophoresis for immunodetection of total TOP1-DPCs and ubiquitylated TOP-DPCs using specific antibodies. In addition, 2 µg of each sample was subjected to slot-blot for immunoblotting with anti-dsDNA antibody to confirm equal DNA loading.

## His pull-down assay
HCT116 cells are washed with 1× PBS and incubated with 220 µl IP lysis buffer (5 mM Tris-HCl pH 7.4, 150 mM NaCl, 1 mM EDTA, 1% NP-40, 0.2% Triton X-100, 5% glycerol, 1 mM DTT, 20 mM N-ethylmaleimide and protease inhibitor cocktail) on a shaker for 15 min at 4 °C, followed by sonication and centrifugation. Supernatant was collected and treated with 1 µl benzonase (250 units/µl) for 1 h. An aliquot (20 µl) of the lysate of each treatment group was saved as input. Lysates were then resuspended in 900 µl Buffer A (6 M guanidine-HCL, 0.1 M $Na_2HPO4/NaH_2PO4$, 10 mM imidazole pH 8.0) containing 100 µl equilibrated Ni-NTA-agarose and rotated overnight at 4 °C. Ni-NTA resin was spun down and washed with TI buffer two times (25 mM Tris HCL, 20 mM imidazole, pH 6.8), followed by resuspension in 2× Laemmli buffer for SDS-PAGE and immunoblotting with various antibodies as indicated.

## FLAG and Myc immunoprecipitation (IP)
HCT116 cells are washed with 1× PBS and incubated with 220 µl IP lysis buffer (5 mM Tris-HCl pH 7.4, 150 mM NaCl, 1 mM EDTA, 1% NP-40, 0.2% Triton X-100, 5% glycerol, 1 mM DTT, 20 mM N-ethylmaleimide and protease inhibitor cocktail) on a shaker for 15 min at 4 °C, followed by sonication and centrifugation. Supernatant was collected and treated with 1 µl benzonase (250 units/µl) for 1 h. An aliquot (20 µl) of the lysate of each treatment group was saved as input. Lysates were resuspended in 900 µl IP lysis buffer containing 2.5 µl anti-FLAG M2 or anti-Myc antibody and rotated overnight at 4 °C. Fifty µl Protein A/G PLUS-agarose slurry was added and incubated with the lysates for another 4 h. After centrifugation, immunoprecipitates were washed with RIPA buffer 2 times then resuspended in 2× Laemmli buffer for SDS-PAGE and immunoblotting with various antibodies as indicated.

## Mass spectrometry
Samples were either separated by SDS-PAGE for in-gel trypsin digestion[67] or in-solution digested with trypsin following the filter-aided sample preparation (FASP) protocol as previously described[68]. Dried peptides were solubilized in 2% acetonitrile, 0.5% acetic acid, 97.5% water for mass spectrometry analysis. They were trapped on a trapping column and separated on a 75 µm × 15 cm, 2 µm Acclaim PepMap reverse phase column (Thermo Scientific) using an UltiMate 3000 RSLCnano HPLC (Thermo Scientific). Peptides were separated at a flow rate of 300 nl/min followed by online analysis by tandem mass spectrometry using a Thermo Orbitrap Fusion mass spectrometer. Peptides were eluted into the mass spectrometer using a linear gradient from 96% mobile phase A (0.1% formic acid in water) to 55% mobile phase B (0.1% formic acid in acetonitrile). Parent full-scan mass spectra were collected in the Orbitrap mass analyzer set to acquire data at 120,000 FWHM resolution; ions were then isolated in the quadrupole mass filter, fragmented within the HCD cell (HCD normalized energy 32%, stepped ±3%), and the product ions analyzed in the ion trap. Proteome Discoverer 2.2 (Thermo) was used to search the data against human proteins from the UniProt database using SequestHT. The search was limited to tryptic peptides, with maximally two missed cleavages allowed. Cysteine carbamidomethylation was set as a fixed modification, and methionine oxidation set as a variable modification. Diglycine modification to lysine was set as a variable modification for experiments to identify sites of enzymatic PTMs. The precursor mass tolerance was 10 ppm, and the fragment mass tolerance was 0.6 Da. The Percolator node was used to score and rank peptide matches using a 1% false discovery rate.

## Recombinant proteins
Human TOP1 was purified from baculovirus as described[69]. Recombinant human ubiquitin was purchased from R&D Systems (Cat. # U-100H). Recombinant human ubiquitin-activating enzyme E1 (UBE1) was purchased from R&D systems (Cat. # E-305). Recombinant human

UbcH5a was purchased from R&D systems (Cat. # E2-616). For purification of the CUL4A-RBX1 complex, synonymously mutated His-Cul4a was co-expressed with His-Rbx1Δ1–14 in SF9 cells. Cells were resuspended in buffer containing 50 mM Tris, 200 mM NaCl, 0.1% Triton X-100, 1 mM TCEP, and 1x SigmaFAST Protease Inhibitors, pH 8.0. Cell suspension was sonicated for 12 cycles at 10 s per cycle, with 30 s of cooling between cycles at 4 °C. Lysate was centrifuged at $20,000 \times g$ for 30 min, and incubated with Ni-NTA agarose for 1 h with rotation at 4 °C. Beads were washed with wash buffer (50 mM Tris, 200 mM NaCl, 10 mM imidazole, 1 mM TCEP, pH 8.0) and eluted with elution buffer (50 mM Tris, 200 mM NaCl, 300 mM imidazole, 1 mM TCEP, pH 8.0). Eluate was injected onto a Superdex200 Increase 10/300 column (GE Healthcare), and the protein complex eluted at 13.8 ml. Protein was concentrated and stored at −80 °C in 50 mM Tris, 200 mM NaCl, 1 mM TCEP, pH 8.0.

## Generation of TOP1cc for its in vitro ubiquitylation

56-nt DNA oligo (TOP1 suicide substrate) with sequence GTCTGTCCGCT-T(biotin)-TAGCGGACAGACATCA-TATCTTCAACGTTTACGTTGAAGATATG was purchased from IDT and annealed in 10 mM Tris-HCl, pH 7.5, 50 mM NaCl and 1 mM EDTA. The DNA substrate is combined with human TOP1 at equal ratio in 10 mM Tris-HCl, pH 7.5, 50 mM KCl, 5 mM $MgCl_2$ 0.1 mM EDTA, and 15 μg/ml BSA at 4 °C overnight. Ten μl in vitro ubiquitylation assay reactions in 1× ubiquitin conjugation reaction buffer (R&D systems Cat. # B-70) contains 10 mM $Mg^{2+}$-ATP solution pH 7.0 (R&D systems Cat. # B-20), protease inhibitor cocktail, 100 nM TOP1-DPC, 10 μM ubiquitin, 50 nM ubiquitin E1, 0.1 μM UbcH5a and CUL4A-RBX1 of indicated concentrations. Reactions were incubated at 37 °C for 30 min, followed by SDS-PAGE and immunoblotting with anti-ubiquitin antibody.

## TOP1-DPC immunofluorescence

TOP1-DPC immunofluorescence was performed as described[45] with slight modification. HCT116 cells grown on chamber slides were treated with SN38 in absence or presence of indicated inhibitors. After inhibitor treatments, cells were washed with PBS and fixed for 15 min at 4 °C in 4% paraformaldehyde in PBS and permeabilized with 0.25% Triton X-100 in PBS for 15 min at 4 °C. The samples were incubated in 2% SDS at room temperature for 10 min, washed and blocked with PBS containing 0.01% Triton X-100, 0.05% Tween 20 and 1% bovine serum albumin (PBSTT-1%BSA). After reaction overnight with TOP1-DPC antibody (Millipore Sigma) in PBSTT-BSA at 4 °C, cells were rinsed with PBSTT and incubated with Alexa Fluor 568-conjugated secondary antibody (Invitrogen) at 1:1000 in PBSTT-BSA for 1 h in subdued light; washed and mounted using mounting medium with DAPI (Vectashield). Images were captured on an instant structured illumination microscope, processed and analyzed using ImageJ.

## DCAF13 immunofluorescence

In total, 100 μM IdU ± 10 μM PEV or ± 1 μM APH were added to DCAF13-FLAG overexpressing HCT116 cells 1 h before CPT (1 μM) treatment. Cells were collected 30 min after CPT treatment and washed with PBS, followed by fixation with 4% PFA for 15 min at room temperature. Cells were then incubated with 1.5 M HCl for 30 min at room temperature for DNA denaturation, followed by permeabilization with 0.25% Triton X-100 in PBS (PBST). Cells were blocked with 1% BSA in 0.1% PBST for 30 min, followed by incubation with rabbit anti-FLAG antibody and mouse anti-BrdU antibody overnight at 4 °C. The next day, Alexa Fluor 568-conjugated anti-rabbit 2nd antibody and Alexa Fluor 488-conjugated anti-mouse antibody were added to the chamber slide for 1 h at room temperature. The slides were incubated with DAPI and mounted using ProLong™ antifade mountant. Images were visualized under a customized instant structured illumination microscope (iSIM). Statistical analysis was performed by ThunderStorm, an ImageJ plugin.

## Neutral comet assay

Neutral comet assays were performed using previously described protocol with minor modifications[35]. In brief, HCT116 cells were treated 20 μM SN38 for 2 h. Cells were then collected for neutral comet assays using the CometAssay Kit (R&D Systems, Catalog # 4250-050-K) following manufacturer's instructions. Images were captured using BioSpa Live Cell Analysis System (Biotek) and tail moment was calculated using OpenComet[70], a plugin for the image processing program ImageJ.

## Proximity ligation assay (PLA)

Duolink PLA fluorescence assay (Sigma-Aldrich, Cat# DUO92101) was performed following manufacturer's instruction. In brief, HCT116 cells were seeded on coverslips and treated with CPT or ETP. After inhibitor treatment, cells were washed with PBS and fixed for 15 min at 4 °C in 4% paraformaldehyde in PBS and permeabilized with 0.25% Triton X-100 in PBS for 15 min at 4 °C. The coverslips were blocked with Duolink blocking solution and incubated with indicated antibodies in the Duolink antibody diluent overnight, followed by incubation with PLUS and MINUS PLA probes, ligation and amplification. Coverslips were then washed and mounted with using mounting medium with DAPI. Images were captured on wide field microscope, processed using ImageJ and analyzed using Imaris.

## iPOND (isolation of proteins on nascent DNA) assay

iPOND was performed as described in Cyril Ribeyre et al.[71] with slight modifications. In brief, $1 \times 10^8$ HCT116 cells were seeded overnight, followed by addition of EdU (10 μM) in presence or absence of CPT (1 μM) for 10 min. Cells were collected and incubated with 2% formaldehyde in PBS buffer for crosslinking for 10 min, followed by incubation with 1.25 M glycine to quench the crosslinking. Cells were subjected to permeabilization in permeabilization buffer (0.25% Triton X-100 in PBS) for 30 min on ice. Cells were then subjected to click reaction in click reaction cocktail (10 mM sodium ascorbate, 2 mM $CuSO_4$, 10 μM biotin-azide (Thermo Fisher, cat# B10184)) for 1 h at 4 °C. Cells were resuspended in lysis buffer (1% SDS in 50 mM Tris-HCl pH 8.0, protease inhibitor cocktail (Thermo Fisher, cat# 78429)). Lysates were sonicated for 20 s for 10 times (40 s cooling down in between). After centrifugation, supernatant was filtered through 100 μm mesh cell strainer, followed by pull-down with streptavidin-magnet beads for 1 h. Thirty μl of lysate were saved as input. The beads were washed with lysis buffer and boiled in SDS sample buffer for immunoblotting.

## Statistical analyses

Three biological repeats were conducted for Figs. 3b, d and 5c, e and Supplementary Fig. 4a. Two biological repeats were conducted for the rest of the immunoblotting experiments unless otherwise stated. Error bars on bar graphs represent standard deviation (SD) or standard error of the mean (SEM) and the $p$ value was calculated using two-tailed Student's $t$ test or two-way ANOVA or for independent samples.

## Reporting summary

Further information on research design is available in the Nature Portfolio Reporting Summary linked to this article.

## Data availability

Uncropped blots are provided in Source Data file, which has been deposited into figshare: https://doi.org/10.6084/m9.figshare.23408060.v1. RNA-seq data have been deposited into dbGaP under access number phs003257.v1.p1. The dataset is under restricted access in a DAS for patient privacy laws. For research purposes, please contact pommier@nih.gov to obtain the data immediately without requiring approval from the DAC. Proteomic data have been deposited into

MassIVE (MSV000091847, https://doi.org/10.25345/C5RN30J25). All remaining data are available in the Article, Supplementary and Source Data files. Source data are provided with this paper.

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

## Acknowledgements

This study was in part supported by the Center for Cancer Research, the Intramural Program of the NCI (Z01 BC 006150) to Y.P., and by the NCI Center for Cancer Research Excellence in Postdoctoral Research Transition Award and the NCI K99 Pathway to Independence Award (1K99 CA 273171) to Y.S.

## Author contributions

Y.S. designed the study. Y.S. wrote the manuscript with input from S.A.B., S.K. and C.J.T. Y.S., S.A.B., X.Z., S.K., V.M.F., Y.A., C.H.C., K.O., A.P., B.W., Y.P.S., J.C., T.T., S.N.H., E.B., Z.I., and C.M. performed the experiments. Specifically, Y.S. performed and analyzed the molecular biology and biochemistry experiments. S.A.B., V.M.F., and C.H.C. performed and analyzed the work in mice. S.K., Y.A., and B.W. performed and analyzed the work including the RNA-Seq in organoids. X.Z., E.B., Z.I., C.M., and C.J.T. performed and analyzed the high-throughput screening. C.X., N.R., D.N., W.D.F., P.S.M., J.C.Y., and C.J.T. provided reagents and/or intellectual input. Y.P. supervised the study and revised the manuscript.

## Funding

## Competing interests

The authors declare no competing interests.

## Additional information

[1]Developmental Therapeutics Branch, Center for Cancer Research, National Cancer Institute, National Institutes of Health, Bethesda, MD 20892, USA. [2]Division of Preclinical Innovation, National Center for Advancing Translational Sciences, National Institutes of Health, Rockville, MD 20850, USA. [3]Genitourinary Malignancies Branch, Center for Cancer Research, National Cancer Institute, National Institutes of Health, Bethesda, MD 20892, USA. [4]Surgery Branch, Center for Cancer Research, National Cancer Institute, National Institutes of Health, Bethesda, MD 20892, USA. [5]Genetics Branch, Center for Cancer Research, National Cancer Institute, National Institutes of Health, Bethesda, MD 20892, USA. [6]Advanced Imaging and Microscopy Resource, National Institute of Biomedical Imaging and Bioengineering, National Institutes of Health, Bethesda, MD 20892, USA. [7]Department of Internal Medicine, University of Texas Southwestern Medical Center, Dallas, TX 75390, USA. [8]Lymphoid Malignancies Branch, Center for Cancer Research, National Cancer Institute, National Institutes of Health, Bethesda, MD 20892, USA. [9]Thoracic and GI Malignancies Branch, Center for Cancer Research, National Cancer Institute, National Institutes of Health, Bethesda, MD 20892, USA. ✉e-mail: yilun.sun@nih.gov; pommier@nih.gov

