## [Peer Review File · Nature Communications]

Reviewers' Comments:

Reviewer #1:

Remarks to the Author:

In this manuscript, the authors identified pevonedistat as synergistically cytotoxic with TOP1 inhibitors in human CRC cells. Mechanically, they reported that CRL4-DCAF13 E3 ligase-mediated ubiquitination and degradation of TOP1-DPCs played a role in the repair of TOP1-DPCs associated with DNA replication. In general, the discoveries the authors present are exciting. However, there are some issues have to be addressed.

Major points:

1. The combination toxicity is the major limitation of the application of drug combination. However, the author didn't intend to rationally evaluate the potential toxicity of drug combination, such as the side effect on liver and kidney function, acute toxicity and the investigation of maximum tolerated dose.
2. Were the data of Fig. 2c and Extended Fig. 2b from the same batch of animal experiments treated with irinotecan and pevonedistat? The data shown on these two figures did not seem to be the same one. In addition, the variation of tumor growth within the group appears to be large according to the data presented.
3. The authors are suggested to compare the difference of mouse tumor weight to show the effect of drug combination.
4. The expression of TOP1-DPCs in the mice tumors (control group vs drug treatment group) should be detected.
5. The authors claimed that DCAF13 was involved in the regulation of TOP1-DPCs elimination, which facilitated the repair of TOP1-DPCs. Are the DCAF13 KO/KD CRC cells more sensitive to the treatment of TOP1 inhibitors?
6. DCAF13 functions as the specific receptor protein of CRL4 complex and directly binds to the substrate. In order to strengthen the conclusion of this work, the authors are suggested to performed GST pull down assay to determine whether DCAF13 binds to TOP1-DPCs directly.
7. How CRL4 is activated by neddylation in response to TOP1-DPCs? Could the enzymes of neddylation (NAE1/UBA3/UBE2M) be activated or regulated in response to TOP1-DPCs?
8. In Fig. 4e-f, Neddylation of CUL4A and CUL4B was stimulated by CPT treatment within 30 minutes. Was the stimulation specific for CRL4 or caused by the activation of neddylation pathway? What about other cullins neddylation?
9. In Line 329, "Fig. 3C, D" is incorrect. Maybe it should be "fig 5c-f". Herein, the authors showed that CRL4 overexpression promoted the removal of TOP1-DPCs. The authors are suggested to determine the effect of CUL4A/4B neddylation-deficient mutants on the removal of TOP1-DPCs.
10. The authors showed that CRL4-DCAF13 regulated the clearance of TOP1-DPCs. Will those tumors with higher activation of Neddylation-CRL pathway be more likely to develop resistance to TOP1 inhibitors? Could the authors compare the drug sensitivity in different tumor cell lines with different expression of CUL4A/4B or DCAF13?

Minor points:

1. In Fig. 3b and the corresponding legend, the authors described the statement "4-hour SN38 treatment". However, in the manuscript, line 221, the authors described "a 2-hour SN38 treatment". The authors should do a double check.
2. In the legend of Fig. 5a, the authors clearly pointed out that CRL4A means the overexpression of Myc-CUL4A plus RBX1-FLAG plasmids. However, in Fig. 5b and some other figures (such as fig. 6c), the authors only described the transfection of Myc-CUL4A in the legend but showed CRL4A in

the figure. In general, CRL represents the complex of Cullin-RING ligase. All these descriptions should be more clearly described.

3. Line 348, "CRLA" was mis-written.

Reviewer #2:

Remarks to the Author:

In their work, the authors have encountered a drug synergism between topoisomerase I inhibition by irinotecan and inhibition of neddylation by pevonedistat in CRC cell lines and patient-derived metastatic CRC organoids. This drug combination also led to a reduced tumor volume and increased survival probability in a nude mice xenotransplantation model. In a series of well-designed and technically sound biochemical experiments, the CUL4-RBX1 complex has been identified responsible for proteasomal degradation and repair of neddylated TOP1-DPCs in a replication-dependent manner. Furthermore, DCAF13 was found in this work as a substrate adaptor to bring TOP1-DPCs and the CRL4 E3-ligase together, thereby mediating TOP1-DPC polyubiquitination and degradation. The mechanistic part of the manuscript regarding the molecular events of neddylation-mediated degradation of TOP1-DPCs via CRL4, which achieves its removal from seDSBs thereby allowing for HR repair, represents a novel and experimentally well-addressed finding very relevant to the field. However, the added value of dual IRI/PEV treatment of liver metastatic CRC and its biological effect on chromosomally instable CRC cells remains unclear, and the provided data do not fully justify the conclusions drawn by the authors on this aspect. In order to provide adequate evidence which should clearly demonstrate the benefit of combined IRI/PEV treatment over current chemotherapeutic strategies against CRC, additional experiments should be performed as specified below.

Major points:

- 1.) The four CRC cell lines HCT116, KM12, SW48, and HCT15 used to demonstrate the synergism between IRI and PEV all represent the microsatellite instable strata of CRC, which is mostly found in right-sided disease and this type of CRC is less prone to form distant metastasis. Since the authors sought to identify new drug combinations effective against liver metastatic CRC, the choice of these CRC cell lines seems suboptimal. The authors should experimentally address the drug-responses on MSI-negative, CIN CRC cell lines (e.g. HT29, SW480/SW620 or T84) and compare drug sensitivity and the IRI/PEV synergism to what has been found in MSI CRC cell lines in order to strengthen the manuscript conclusions.
- 2.) While IRI/PEV synergism has been confirmed in liver metastatic CRC organoids, no information on the microsatellite instability status or the driver gene mutational spectrum (APC, KRAS/NRAS/BRAF, PIK3CA, TP53, SMADs) of these lines has been provided. The authors should include these data in their manuscript.
- 3.) While xenograft tumors in the vehicle and monotherapy groups display similar tumor volumes after the 21 day treatment period, animals treated with PEV and IRI alone seem to benefit substantially with respect to the survival probability. The authors should provide a biological explanation for this phenomenon. Especially, the drug-response of single or combined treatments has been studied only on the level of cell viability (ex vivo) or tumor volume (in vivo). It would be very informative to analyze and compare the percentage of proliferating (EdU in vitro and MKI67 in vivo), cell cycle arrested (EdU in vitro, CDKN1A/p21 in vivo), and apoptotic (Annexin V in vitro, and cleaved caspase-3 in vitro and in vivo) CRC cells between single and combination treatments. Is the fraction of arrested and/or apoptotic cells in tumors increased upon IRI/PEV dual treatment relative to monotherapies? Is this effect dependent on functional or mutant p53? (see also point 4).
- 4.) The authors should provide new data which answer these questions.
- 4.) The tumor cellular response to cytotoxic agents is influenced by the p53/p21 pathway, and activation of p53, whose functionality is lost in the majority of CRC and especially in the highly metastatic CIN fraction, might act chemo-protective by favoring cell cycle arrest and DNA repair over apoptosis. Another study on MLN4924 (PEV) has demonstrated that p53-negative CRC cells react more sensitive to this drug: Notably, apoptosis seems to play a more important role in the anti-proliferative effect of PEV when compared to arrest/senescence, which is triggered by PEV preferentially in a p53 wild-type background (Lin JJ, Milhollen MA, Smith PG, Narayanan U, Dutta A. NEDD8-targeting drug MLN4924 elicits DNA rereplication by stabilizing Cdt1 in S phase,

triggering checkpoint activation, apoptosis, and senescence in cancer cells. *Cancer Res.* 2010 Dec 15;70(24):10310-20). For that reason, HCT116 cells do not represent an appropriate model to study the in vivo synergism/benefit of combined IRI/PEV treatment relative to monotherapy in metastatic CRC. In vivo treatment experiments should be carried out either on xeno-transplanted liver metastatic CRC organoids negative for p53, or on p53-negative, chromosomally instable CRC cell line-derived tumors.

5.) Although the experimental data depicted in Figures 3-6 provide new, very clear, experimentally well-performed, and concise insights into the molecular mechanism of TOP1-DPCs ubiquitylation by CRL4 during DNA replication, and also shows its overall importance in the context of irinotecan-mediated DNA damage, all experiments have been performed exclusively in p53 wild-type HCT116 cells. Inhibition of neddylation might also affect the activity of Mdm2/p53/p21 signaling (Xirodimas DP, Saville MK, Bourdon JC, Hay RT, Lane DP. Mdm2-mediated NEDD8 conjugation of p53 inhibits its transcriptional activity. *Cell.* 2004 Jul 9;118(1):83-97), and proteasome inhibition very likely stabilizes and therefore activates p53. This might impact on cell cycle arrest, DNA repair, and thereby also the extent of γ -H2AX abundance after combination treatment. Especially the effects of neddylation- and proteasome-inhibition on SN38-induced seDSBs (as shown in Figure 3f), and on SN38-induced pCHK1, pRPA, and γ -H2AX levels (shown in Figure 3g) should be experimentally addressed also in a p53-null CRC background in order to clarify if they are dependent on or independent of p53 functionality.

6.) Standard first-line chemotherapeutic strategies against liver metastatic CRC include combination treatment with FOLFIRI (5'-FU, folinic acid, irinotecan) rather than irinotecan alone. It would be very informative to see if there is also a synergism between dual FOLFIRI plus PEV treatment in chromosomally instable CRC cells or organoids.

Minor points:

- line 340: "unknow" should be "unknown"
- line 350: "CRL4-pontentiated" should be "CRL4-potentiated"
- line 439: "tissues including colon does not..." should be "tissues including colon do not..."
- line 473: "TOP1-targed therapies" should be "TOP-targeted therapies"

Reviewer #3:

Remarks to the Author:

Professor Yves Pommier's manuscript describes a very intriguing cancer treatment strategy. Overall, the manuscript's writing is imperfect, with poorly organized paragraphs. The figures are not organized properly, and the small font is unreadable. Figures (figures 1d&e; figures 3 d,e&g; figure 4~7) must be reorganized for improved presentation of quantitative data. Some figures should be relocated to the supplementary materials.

Despite this, the topic is intriguing, and significant new discoveries have been made. The reviewer concurs with the authors that Figure 7 is incredibly intriguing. After major issues have been resolved and figures have been reorganized, the manuscript is ready for a thorough evaluation.

Major issues;

1. Irinotecan (IRI) is a vital antitumor medication, but it is rarely administered alone. General treatment guidelines recommend IRI with FOLFIRI (Leucovorin calcium+Fluorouracil) or FOLFOXIRI (additionally Oxaliplatin). In the authors' experiments, the reviewer observed a growth-inhibiting effect, but not a strong cytotoxic effect, under 1:1 synergism of IRI and PEV.

The reviewer recommends that the authors conduct additional experiments involving the combined treatment of PEV with FOLFIRI or FOLFOXIRI. Other combined options are acceptable as well. With strong cytotoxicity in the combined cases, the efficacy of IRI could be increased for a higher clinical value.

2. The reviewer observed the high PEV concentration in PDO experiments, which is generally acceptable when compared to 2D cell line concentrations. However, the reviewer observed a higher PEV concentration under the fixed concentration of SN38 when applied to PDO#3. The

PDO#3 should be distinguished from other PDOs by genetic characteristics not described in the manuscript. Greater NEDD8 expression could be the reason.

The high resistance of PDO#3 must be confirmed through additional experiments in order to bolster the main argument of this manuscript.

Additional information regarding PDOs, such as clinical and genetic donor information, is also required.

3. In Combi Tx cases, cell seeding for organoid formation should have a low confluence, as estimated from images of grown organoids. Fluorescent images with live/dead information or wide-field views with additional organoids would be useful to support the conclusion of figure 2a.

Minor issues;

1. Abbreviations must be defined when they first appear.
2. Figure 3f needs to be replaced.
3. Figure font sizes should be synchronized. The locations of graph legends should be arranged.

Reviewer #4:

Remarks to the Author:

Sun et al, here describe the identification of a potential drug combination strategy for the treatment of colorectal cancer. Via a high-throughput screen of more than 2000 compounds, they identify pevonedistat (PEV) as having some activity against HCT-116, before extending the screen to look for synergistic drug combinations. A handful of combination were identified with TOP1 being the most notable given its use as a first and second-line treatment in metastatic CRC. Validation of this combination was conducted in other model systems and the authors went on to show the mechanism and pathway by how which this combination works with implications not only for CRC but for other cancers where TOP1 inhibition is used as a treatment.

I support the publication of this manuscript and I have no major comments/reviews. I do feel the paper would read better if the following could be incorporated:

- why was HCT-116 the CRC cell line screened? Provide information on the genomics of this model?
- With respect to validation of the combination, was MSI and MSS disease accounted for. Comment on this in the paper.
- What were the genomics of the 3 PDO's used? Had the patients received a TOP1 inhibitor? Did they respond like the PDO?
- Comment on use of this combination in patients who are resistant to TOP inhibitors.
- Suggest performing an apoptosis assay on cell lines and organoids to determine if in vitro you observe any cell kill or whether combination is growth-inhibitory only.

Reviewer #1 - Neddylation, ubiquitination - (Remarks to the Author):

In this manuscript, the authors identified pevonedistat as synergistically cytotoxic with TOP1 inhibitors in human CRC cells. Mechanically, they reported that CRL4-DCAF13 E3 ligase-mediated ubiquitination and degradation of TOP1-DPCs played a role in the repair of TOP1-DPCs associated with DNA replication. In general, the discoveries the authors present are exciting. However, there are some issues have to be addressed.

Major points:

1. The combination toxicity is the major limitation of the application of drug combination. However, the author didn't intend to rationally evaluate the potential toxicity of drug combination, such as the side effect on liver and kidney function, acute toxicity and the investigation of maximum tolerated dose.

Answer: Thank you for suggesting toxicity study for assessing the combination. In collaboration with William Douglas Figg Sr., Pharm.D. the Chief of Genitourinary Malignancies Branch, NCI, and his lab members Genitourinary Malignancies Branch, we measured toxicity in mice treated with irinotecan (20 mg/kg) + pevonedistat (2.5 mg/kg) and observed no overt toxicity upon completion of this study (Supplementary figure 2d-f; Supplementary table 3). Specifically:

- No significant weight loss was observed.
- Clinical chemistry, including liver enzymes and renal function biomarkers, were within normal limits between groups and for reference data from Mouse Phenome Database (<https://phenome.jax.org/>).
- CBC –elevations in WBC counts at day 10 and decreased platelets for PEV treated mice normalized compared to other groups when including the day 22 timepoint.
- No significant gross lesions were observed. Gross findings including reactive lymph nodes, reactive gut-associated lymphoid tissue, changes in the lung consistent with agonal changes, and changes in the reproductive tract, which are common background findings in laboratory mice.

2. Were the data of Fig. 2c and Extended Fig. 2b from the same batch of animal experiments treated with irinotecan and pevonedistat? The data shown on these two figures did not seem to be the same one. In addition, the variation of tumor growth within the group appears to be large according to the data presented.

Answer: The data of Fig. 2c and Supplementary Fig. 2b are indeed from the same animal experiment. Fig. 2c depicts relative tumor growth (fold change of each individual tumor with respect to its initial tumor volume), while Supplementary Fig. 2b shows the mean tumor volume of each treatment group. The variation of tumor growth we believe is within the range of biological variability.

3. The authors are suggested to compare the difference of mouse tumor weight to show the effect of drug combination.

Answer: Thank you for the suggestion. We do have data for tumor weight and have included those data as Supplementary figure 2c.

4. The expression of TOP1-DPCs in the mice tumors (control group vs drug treatment group) should be detected.

Answer: We performed ICE assay using the tumor tissues and found that the combination group exhibited accumulation of TOP1-DPCs compared to the irinotecan-treated sample (now supplementary figure 3b).

5. The authors claimed that DCAF13 was involved in the regulation of TOP1-DPCs elimination, which facilitated the repair of TOP1-DPCs. Are the DCAF13 KO/KD CRC cells more sensitive to the treatment of TOP1 inhibitors?

Answer: We have generated a DCAF13 CRISPR KO clone in HEK293 human embryonic kidney cells and found increased sensitivity to SN38 (supplementary figure 7c).

6. DCAF13 functions as the specific receptor protein of CRL4 complex and directly binds to the substrate. In order to strengthen the conclusion of this work, the authors are suggested to performed GST pull down assay to determine whether DCAF13 binds to TOP1-DPCs directly.

Answer: In consistence with figure 6e, we performed reciprocal IP assay using an antibody targeting endogenous DCAF13 and found that DCAF13 directly binds to TOP1-DPCs upon their formation by SN38 using an antibody specifically targeting TOP1-DPC (PMID: 26917015) (supplementary figure 6d).

7. How CRL4 is activated by neddylation in response to TOP1-DPCs? Could the enzymes of neddylation (NAE1/UBA3/UBE2M) be activated or regulated in response to TOP1-DPCs?

Answer: Thank you for this interesting question. In order for neddylation to take place, the neddylation system (E1, E2 and E3) must be activated. Yet, there is no marker for their own activation other than measurement of neddylation of their substrate (CRL4 in our case).

8. In Fig. 4e-f, Neddylation of CUL4A and CUL4B was stimulated by CPT treatment within 30 minutes. Was the stimulation specific for CRL4 or caused by the activation of neddylation pathway? What about other cullins neddylation?

Answer: As suggested, we have assessed CUL3 neddylation and found that CUL3 is also neddylated in response to TOP1cc induction by SN38. Please see the figure below. This is consistent with Figure 4a showing that upregulation of CUL4 + RBX increased TOP1-DPC ubiquitylation, albeit not significant (Supplementary figure 4b). We hypothesize that CUL3 might be involved in transcription-coupled ubiquitylation of TOP1-DPCs and plays more important role TOP1-DPC repair in cancers highly dependent on transcription other than CRC. We feel that studying the implication of CUL3 deserves further investigation, and we plan to explore it as an independent project. As upregulation of other cullin members did not simulate TOP1-DPC ubiquitylation (Figure 4a), we conclude that the neddylation stimulation is specific for CRL4. This point has been included in the Discussion of our revised manuscript.

Figure legend: HCT116 cells were transfected with the indicated plasmids (CUL3 WT-FLAG, CUL3 K712R (neddylation-deficient)-FLAG or HA-NEDD8) with SN38 (10 μ M) +/- PEV treatment (μ M) for 30 min. Samples were immunoprecipitated with FLAG antibody IP and WB with HA antibody and FLAG antibody.

9. In Line 329, “Fig. 3C, D” is incorrect. Maybe it should be “fig 5c-f”. Herein, the authors showed that CRL4 overexpression promoted the removal of TOP1-DPCs. The authors are suggested to determine the effect of CUL4A/4B neddylation-deficient mutants on the removal of TOP1-DPCs.

Answer: Thank you for noting this editorial error. We have revised the text accordingly.

As suggested, we have evaluated the effect of CUL4A/4B neddylation-deficient mutants on the removal of TOP1-DPCs and found that they are unable to remove TOP1-DPCs compared their WT counterparts (Figure 4g).

10. The authors showed that CRL4-DCAF13 regulated the clearance of TOP1-DPCs. Will those tumors with higher activation of Neddylation-CRL pathway be more likely to develop resistance to TOP1 inhibitors? Could

the authors compare the drug sensitivity in different tumor cell lines with different expression of CUL4A/4B or DCAF13?

Answer: We have chosen 7 cancer cell lines with different CUL4 expressions (Supplementary figure 4g) and performed WB to measure their levels. We next performed drug sensitivity assays and observed that cell lines with higher levels of CUL4 are in general more resistant to SN38 than those expressing lower levels of CUL4 (supplementary 4h). These new data have been included in our revised manuscript.

Minor points:

1. In Fig. 3b and the corresponding legend, the authors described the statement “4-hour SN38 treatment”. However, in the manuscript, line 221, the authors described “a 2-hour SN38 treatment”. The authors should do a double check.

Answer: Thank you; We have corrected the manuscript accordingly.

2. In the legend of Fig. 5a, the authors clearly pointed out that CRL4A means the overexpression of Myc-CUL4A plus RBX1-FLAG plasmids. However, in Fig. 5b and some other figures (such as fig. 6c), the authors only described the transfection of Myc-CUL4A in the legend but showed CRL4A in the figure. In general, CRL represents the complex of Cullin-RING ligase. All these descriptions should be more clearly described.

Answer: As suggested, all the description have been clarified.

3. Line 348, “CRLA” was mis-written.

Answer: Thank you; We have corrected the manuscript accordingly.

Reviewer #2 - CRC organoids (Remarks to the Author):

In their work, the authors have encountered a drug synergism between topoisomerase I inhibition by irinotecan and inhibition of neddylation by pevonedistat in CRC cell lines and patient-derived metastatic CRC organoids. This drug combination also led to a reduced tumor volume and increased survival probability in a nude mice xenotransplantation model. In a series of well-designed and technically sound biochemical experiments, the CUL4-RBX1 complex has been identified responsible for proteasomal degradation and repair of neddylated TOP1-DPCs in a replication-dependent manner. Furthermore, DCAF13 was found in this work as a substrate adaptor to bring TOP1-DPCs and the CRL4 E3-ligase together, thereby mediating TOP1-DPC polyubiquitination and degradation. The mechanistic part of the manuscript regarding the molecular events of neddylation-mediated degradation of TOP1-DPCs via CRL4, which achieves its removal from seDSBs thereby allowing for HR repair, represents a novel and experimentally well-addressed finding very relevant to the field. However, the added value of dual IRI/PEV treatment of liver metastatic CRC and its biological effect on chromosomally unstable CRC cells remains unclear, and the provided data do not fully justify the conclusions drawn by the authors on this aspect. In order to provide adequate evidence which should clearly demonstrate the benefit of combined IRI/PEV treatment over current chemotherapeutic strategies against CRC, additional experiments should be performed as specified below.

Major points:

1.) The four CRC cell lines HCT116, KM12, SW48, and HCT15 used to demonstrate the synergism between IRI and PEV all represent the microsatellite unstable strata of CRC, which is mostly found in right-sided disease and this type of CRC is less prone to form distant metastasis. Since the authors sought to identify new drug combinations effective against liver metastatic CRC, the choice of these CRC cell lines seems suboptimal. The authors should experimentally address the drug-responses on MSI-negative, CIN CRC cell lines (e.g. HT29, SW480/SW620 or T84) and compare drug sensitivity and the IRI/PEV synergism to what has been found in MSI CRC cell lines in order to strengthen the manuscript conclusions.

Answer: Thank you for the suggestion. We performed drug sensitivity assays in HT29 and SW620 CRC cell lines and found that pevonedistat sensitized those cells to SN38 (Supplementary figure 1a), suggesting that the IRI/PEV synergism is independent on the status of microsatellite instability and chromosome instability.

2.) While IRI/PEV synergism has been confirmed in liver metastatic CRC organoids, no information on the microsatellite instability status or the driver gene mutational spectrum (APC, KRAS/NRAS/BRAF, PIK3CA, TP53, SMADs) of these lines has been provided. The authors should include these data in their manuscript.

Answer: Thank you for the suggestion. The primary therapeutic mechanism of TOP1 inhibitor is the induction of TOP1-DPCs, and our molecular analyses show that the IRI/PEV synergism stems from the defective repair of TOP1-DPCs regulated by neddylation, which is independent on microsatellite instability status or the driver gene mutational spectrum (as evidenced by our new experiment showing that p53 does not play a role in TOP1-DPC repair regardless of neddylation status: supplementary figure 3c). We therefore decided not to determine the microsatellite instability status or the driver gene mutational spectrum of the CRC organoids and to focus on TOP1-DPC degradation; and will be reported in a subsequent study by our collaborators who established the organoids.

3.) While xenograft tumors in the vehicle and monotherapy groups display similar tumor volumes after the 21day treatment period, animals treated with PEV and IRI alone seem to benefit substantially with respect to the survival probability. The authors should provide a biological explanation for this phenomenon. Especially, the drug-response of single or combined treatments has been studied only on the level of cell viability (ex vivo) or tumor volume (in vivo). It would be very informative to analyze and compare the percentage of proliferating (EdU in vitro and MKI67 in vivo), cell cycle arrested (EdU in vitro, CDKN1A/p21 in vivo), and apoptotic (Annexin V in vitro, and cleaved caspase-3 in vitro and in vivo) CRC cells between single and combination treatments. Is the fraction of arrested and/or apoptotic cells in tumors increased upon IRI/PEV dual treatment relative to monotherapies? Is this effect dependent on functional or mutant p53? (see also point 4). The authors should provide new data which answer these questions.

Answer: The average tumor volume of the vehicle group is over 1,300 mm³ whereas that of the PEV group is below 1,000 mm³ and that of the IRI group 700 mm³. The differences are significant.

As suggested, we have performed EdU FACS and observed that PEV reversed cell cycle (S phase) arrest by SN38 (Figures below). This is because neddylation inhibition induces DNA re-replication by preventing CRL4-mediated degradation of CDT1 (PMID: 21159650), a replication licensing factor whose degradation is essential in preventing re-replication. We have also included experiments showing that SN38 induces the degradation of CDT1 in response to TOP1-DPCs, and that PEV prevented the degradation of CDT1. For this observation is not directly linked to the repair of TOP1-DPCs, we decided not to include those results in the current manuscript, and plan to carefully investigate it as a subsequent independent project.

Figure legend: a) Biophasic analysis of SN38 and PEV on replication. Cell cycle evaluation in HT29 cells following exposure to SN38 (1 μM), PEV (1 μM) or the combination by flow cytometry analysis. Cells were pulse-labeled with 50 μM 5-ethynyl-2'-deoxyuridine (EdU) for 30 min before adding PEV (1h pre-treatment) then SN38 for 1h. Cells were harvested 18 hour-post SN38. EdU and DAPI staining were detected by flow cytometry. **b)** monophasic analysis of SN38 and PEV on replication. Cell cycle effects were determined by DAPI staining. **c)** WB in HT29 cells showing that PEV (2 μM) prevents CDT1 degradation under normal conditions and upon exposure to SN38.

4.) The tumor cellular response to cytotoxic agents is influenced by the p53/p21 pathway, and activation of p53, whose functionality is lost in the majority of CRC and especially in the highly metastatic CIN fraction, might act chemo-protective by favoring cell cycle arrest and DNA repair over apoptosis. Another study on MLN4924 (PEV) has demonstrated that p53-negative CRC cells react more sensitive to this drug: Notably, apoptosis seems to play a more important role in the anti-proliferative effect of PEV when compared to arrest/senescence, which is triggered by PEV preferentially in a p53 wild-type background (Lin JJ, Milhollen MA, Smith PG, Narayanan U, Dutta A. NEDD8-targeting drug MLN4924 elicits DNA rereplication by stabilizing Cdt1 in S phase, triggering checkpoint activation, apoptosis, and senescence in cancer cells. *Cancer Res.* 2010 Dec 15;70(24):10310-20). For that reason, HCT116 cells do not represent an appropriate model to study the in vivo synergism/benefit of combined IRI/PEV treatment relative to monotherapy in metastatic CRC. In vivo treatment experiments should be carried out either on xeno-transplanted liver metastatic CRC organoids negative for p53, or on p53-negative, chromosomally unstable CRC cell line-derived tumors.

Answer: Thank you. As suggested, we tested the combination in multiple cell lines and organoids with different p53 status. As p53 does not change the levels of drug-induced TOP1-DPC (Supplementary figure 3c) both in the absence and presence of PEV. The role of p53 in TOP1-induced cell killing is likely downstream of TOP1-DPC repair and therefore warrants independent investigation. Involving experiments on p53 may complicate our current study, which already contains a large number of figures, and is focused on TOP1-DPC repair.

5.) Although the experimental data depicted in Figures 3-6 provide new, very clear, experimentally well-performed, and concise insights into the molecular mechanism of TOP1-DPCs ubiquitylation by CRL4 during DNA replication, and also shows its overall importance in the context of irinotecan-mediated DNA damage, all experiments have been performed exclusively in p53 wild-type HCT116 cells. Inhibition of neddylation might also affect the activity of Mdm2/p53/p21 signaling (Xirodimas DP, Saville MK, Bourdon JC, Hay RT, Lane DP. *Cell.* 2004 Jul 9;118(1):83-97), and proteasome inhibition very likely stabilizes and therefore activates p53. This might impact on cell cycle arrest, DNA repair, and thereby also the extent of γ -H2AX abundance after combination treatment. Especially the effects of neddylation- and proteasome-inhibition on SN38-induced seDSBs (as shown in Figure 3f), and on SN38-induced pCHK1, pRPA, and γ -H2AX levels (shown in Figure 3g) should be experimentally addressed also in a p53-null CRC background in order to clarify if they are dependent on or independent of p53 functionality.

Answer: As suggested, we performed ICE assays in p53 mutant CRC cells (SW620) and did not see any change in the levels of drug-induced TOP1-DPCs. In line with the points raised above, we decided not to include these data to our manuscript and would rather pursue the role of p53 in TOP1-DPC repair separately.

Figure legend: Upper panels: ICE assays in WT and CUL4A/B knocking-down SW620 cells (treated with SN38 (10 μ M), PEV (10 μ M) or the combination for 2hrs) suggesting that CUL4A/B are required for the removal of TOP1-DPCs regardless of p53 mutation status. Lower panel: quantitation of ICE assays including the representative image shown in the upper panel. Asterisks indicate significant difference.

6.) Standard first-line chemotherapeutic strategies against liver metastatic CRC include combination treatment with FOLFIRI (5'-FU, folinic acid, irinotecan) rather than irinotecan alone. It would be very informative to see if there is also a synergism between dual FOLFIRI plus PEV treatment in chromosomally instable CRC cells or organoids.

Answer: We have performed combination treatment of PEV and with FOLFIRI (5'-FU, folinic acid, irinotecan) in our CRC organoid #2 and found PEV at 2.5 uM sensitized the organoid to FOLFIRI (Supplementary figure 2k).

Minor points:

- line 340: “unknow” should be “unknown”
- line 350: “CRL4-pontentiated” should be “CRL4-potentiated”
- line 439: “tissues including colon does not...” should be “tissues including colon do not...”
- line 473: “TOP1-targed therapies” should be “TOP-targeted therapies”

Answer: Thank you for the editorial suggestions , which we have corrected in our revised manuscript.

Reviewer #3 - High-throughput screening - (Remarks to the Author):

Professor Yves Pommier's manuscript describes a very intriguing cancer treatment strategy. Overall, the manuscript's writing is imperfect, with poorly organized paragraphs. The figures are not organized properly, and the small font is unreadable. Figures (figures 1d&e; figures 3 d,e&g; figure 4~7) must be reorganized for improved presentation of quantitative data. Some figures should be relocated to the supplementary materials.

Despite this, the topic is intriguing, and significant new discoveries have been made. The reviewer concurs with the authors that Figure 7 is incredibly intriguing. After major issues have been resolved and figures have been reorganized, the manuscript is ready for a thorough evaluation.

Major issues;

1. Irinotecan (IRI) is a vital antitumor medication, but it is rarely administered alone. General treatment guidelines recommend IRI with FLOFIRI (Leucovorin calcium+Fluorouracil) or FOLFOXIRI (additionally Oxaliplatin). In the authors' experiments, the reviewer observed a growth-inhibiting effect, but not a strong cytotoxic effect, under 1:1 synergism of IRI and PEV. The reviewer recommends that the authors conduct additional experiments involving the combined treatment of PEV with FOLFIRI or FOLFOXIRI. Other combined options are acceptable as well. With strong cytotoxicity in the combined cases, the efficacy of IRI could be increased for a higher clinical value.

Answer: Thank you for finding the topic significant, the constructive criticisms and editorial suggestions. As suggested, we have performed additional combination treatments of PEV with FOLFIRI (5'-FU, folinic acid, irinotecan) showing that PEV sensitized the organoid to irinotecan as well in the FOLFIRI combination (Supplementary figure 2i). Additionally, we wish to stress that a new main finding of our manuscript is the identification of a novel repair pathway for TOP1-DPCs revealing by cullin 4 and the TOP1 adaptor, DCAF13, explaining the effects of the clinical neddylation inhibitor PEV based on its inhibition of cullins.

2. The reviewer observed the high PEV concentration in PDO experiments, which is generally acceptable when compared to 2D cell line concentrations. However, the reviewer observed a higher PEV concentration under the fixed concentration of SN38 when applied to PDO#3. The PDO#3 should be distinguished from other PDOs by genetic characteristics not described in the manuscript. Greater NEDD8 expression could be the reason.

The high resistance of PDO#3 must be confirmed through additional experiments in order to bolster the main argument of this manuscript.

Additional information regarding PDOs, such as clinical and genetic donor information, is also required.

Answer: As suggested, we performed Western blotting control experiments in all three PDOs and found that PDO #3 exhibited higher levels of CUL4 proteins (Supplementary figure 4f), likely resulting in its relative resistance to PEV + IRI combination.

3. In Combi Tx cases, cell seeding for organoid formation should have a low confluence, as estimated from images of grown organoids. Fluorescent images with live/dead information or wide-field views with additional organoids would be useful to support the conclusion of figure 2a.

Answer: All the organoids grow extremely slowly. We do have more images of different magnification:

If needed, we can provide those images as supplementary raw image dataset.

Minor issues;

1. Abbreviations must be defined when they first appear.
2. Figure 3f needs to be replaced.
3. Figure font sizes should be synchronized. The locations of graph legends should be arranged.

Answer: We apologize for these editorial deficiencies and thank you for noting them. As suggested, we have synchronized the font sizes. Because figure 1d & e and figure 3d are exported figures, we cannot change their font sizes. We have provided the link to the data for downloading the original figures.

Reviewer #4 - CRC organoids, drug screening - (Remarks to the Author):

Sun et al, here describe the identification of a potential drug combination strategy for the treatment of colorectal cancer. Via a high-throughput screen of more than 2000 compounds, they identify pevonedistat (PEV) as having some activity against HCT-116, before extending the screen to look for synergistic drug combinations.

A handful of combination were identified with TOP1 being the most notable given its use as a first and second-line treatment in metastatic CRC. Validation of this combination was conducted in other model systems and the authors went on to show the mechanism and pathway by how which this combination works with implications not only for CRC but for other cancers where TOP1 inhibition is used as a treatment.

I support the publication of this manuscript and I have no major comments/reviews. I do feel the paper would read better if the following could be incorporated:

- why was HCT-116 the CRC cell line screened? Provide information on the genomics of this model?

Answer: HCT116 is among the most common CRC cell line used for topoisomerase inhibitor studies. They grow very well for high throughput studies and can be readily modified genetically. Accordingly, we used them for genetically engineering cell lines for mechanistic and translational studies.

- With respect to validation of the combination, was MSI and MSS disease accounted for. Comment on this in the paper.

Answer: Thank you for your suggestion. The primary therapeutic mechanism of TOP1 inhibitors is the induction of TOP1-DPCs, and the study reveals that the IRI/PEV synergism stems from the inhibition of TOP1-DPCs repair regulated by neddylation. We therefore believe that determination of the microsatellite instability status of the CRC organoids is beyond the scope of the present study; and will be reported in a subsequent study by our collaborators who established the organoids.

- What were the genomics of the 3 PDO's used? Had the patients received a TOP1 inhibitor? Did they respond like the PDO?

Answer: These points have been included in our revised manuscript.

Patients 1 and 2 did receive TOP1 inhibitor treatment. Below is the clinical information:

CRC PDO #1

2013 Abdominal pain- large sigmoid colon ca. Resected with 1/41 positive LN (stage T3N1Mx). Started adjuvant FOLFOX ×12 cycles.

2016 New liver lesion noted on routine CT surveillance. Resected followed by adjuvant FOLFIRI ×8 cycles. Sequencing showing APC (R213*), TP53 (F270I), ERBB2 amplification, KDM6A (N839fs*27); KRAS/NRAS/BRAF WT, MSS.

2018 Palpable chest wall mass. Completed neoadjuvant FOLFOX ×6 cycles and underwent resection of chest wall mass as well as resection of liver lesions.

2019 Irinotecan + cetuximab for progressive disease in the lungs and chest wall. Then underwent second resection of chest wall mass.

2019 Palliative pertuzumab + trastuzumab due to ERBB2 amplification- NR.

2019 Lonsurf

2020 Palliative radiation to R chest wall.

2020 Lung TIL harvest, (tumor for organoid procured).

2021 PD, unable to receive immunotherapy. Patient died of disease.

CRC PDO #2

2019 Rectal bleeding, lower rectal primary. lung lesions, biopsy positive (stage IV). KRAS G12D mutation. FOLFOX for 5 mo with no response.

2020 FOLFIRI + Avastin: decrease in multiple pulmonary nodules and perirectal lymph node

2020 Progressive disease - two doses of Lonsurf. Local recurrence in rectum.

2020 TIL harvest x2 of lung lesions (Organoid procured from this surgery).

2022 Delayed due to lab closure—disease progressed and pt not treated.

CRC PDO #3

Lymph node, cervical, right, needle core biopsy: Moderately to poorly differentiated metastatic adenocarcinoma consistent with colorectal primary.

- Comment on use of this combination in patients who are resistant to TOP inhibitors.

Answer: As requested, we have commented on the combination in patients who are resistant to TOP1 inhibitors in the Discussion section of our revised manuscript.

- Suggest performing an apoptosis assay on cell lines and organoids to determine if in vitro you observe any cell kill or whether combination is growth-inhibitory only.

Answer: Thank you. We have performed apoptosis assays in two CRC cell lines and found the combination increased cleaved caspase-3 levels compared to SN38 single treatment (Supplementary figure 1b and c), suggesting that neddylation inhibition enhances TOP1 inhibitor-induced apoptotic cell killing.

Reviewers' Comments:

Reviewer #1:

Remarks to the Author:

The authors have extensively addressed my concerns and revised the manuscript based on earlier reviews.

For the point 6 I raised last time, the authors performed an endogenous IP using DCAF13 antibody other than GST pull down assay. They showed that DCAF13 bound to TOP1-DPC (Fig s6d). Since the endogenous IP could not represent a direct binding of two proteins, it is suggested to delete the word "directly" in the revised manuscript (line 350-352).

Reviewer #3:

Remarks to the Author:

1. Organoid issue

The organoids used in this manuscript must be thoroughly validated. The expression of colon and colon cancer markers must be confirmed with the appropriate genetic data.

In drug testing experiments, the organoids used are too small. Some are only 30–40 microns in diameter and contain few cells. The statistical analysis of drug efficacy results seemed not be conducted with the correct number of cells.

The quantity of organoids used in experiments differs from the quantity of controls depicted in the mCRC images in the rebuttal letter.

2. In Figure 2b, the images of PEV and SN38 are cropped from the same image. It appears to be manipulated, for which the authors must provide a convincing explanation. Figure 2a's data is therefore not supported by the images.

The reviewer suspects the remaining data in figure 2a (PDOs #2 and #3). The authors must submit the images used to produce the graphs.

3. The authors explained that the high concentration of PEV in sample #3 was due to the high CUL4 expression.

The levels of CUL4 expression in #1 and #2 are comparable (Supplementary Figure 4), but their viability is very different. These distinctions are also observed in experiments with cell lines (Figure 1e).

In addition to CUL4, it appears that mCRC 3 must process PEV in high concentration for another reason.

The graphs in figure 1e exhibit large variations in the y-axis, which infers the diversity of the cell lines. It would be desirable for WB to display the amount of CUL4 expression in these lines. Do HCT15, KM12, and SW48 express CUL4 at a higher level than 837? SW837 has a vastly superior response.

Comparing the values in Supplemental Figure 1a and Figure 1e, CUL4 expression in HT29 was quite high, and the results were comparable to those in Figure 1e for the cell lines shown. If all the cell lines depicted in Figure 1e have high CUL4 expression levels, the high CUL4 expression in mCRC#3 for the requirement of high C PEV could be supported. If not, another cause may account for the high concentration.

additional issues;

1. There are no loading control (tubulin) data in supplementary figure 4. It is necessary to have a complete blot image of the expression of CUL4 and TOP1 with the same amount quantified.

2. In supplementary Figure 4g & h, the authors must illustrate the difference in SN38 reactivity between lines with high and low CUL4 expression, as well as the combination with PEV.

3. Please check the attached pdf file. Some data were unnecessarily cropped. The reviewer asked the authors to submit original images.

Reviewer #4:

Remarks to the Author:

All of my comments and have been addressed in the revised manuscript and I support the publication of this study.

Reviewer #5:

Remarks to the Author:

The authors have addressed my major concerns appropriately. The novelty and quality of the (new) data and the newly provided clarification of major issues have increased the overall quality of the manuscript substantially. Figure quality and layout also have been improved compared to the first manuscript version.

I recommend the revised manuscript by Sun et al. for publication in Nature Communications.

Reviewer #1 (Remarks to the Author):

The authors have extensively addressed my concerns and revised the manuscript based on earlier reviews.

For the point 6 I raised last time, the authors performed an endogenous IP using DCAF13 antibody other than GST pull down assay. They showed that DCAF13 bound to TOP1-DPC (Fig s6d). Since the endogenous IP could not represent a direct binding of two proteins, it is suggested to delete the word “directly” in the revised manuscript (line 350-352).

Thank you. As suggested, we have deleted the word “directly”.

Reviewer #3 (Remarks to the Author):

1. Organoid issue

The organoids used in this manuscript must be thoroughly validated. The expression of colon and colon cancer markers must be confirmed with the appropriate genetic data.

In collaboration with Dr. Paul Meltzer (Chief of Genetics Branch at NCI) and his lab member Bob Walker, we performed RNA-seq for all the organoids and generated hierarchical clustering heatmap showing differences in RNA expression profiling between the organoids (now supplementary figure 2a). Under the sequencing, a total of 16,784 genes were detected in the samples. The RNA seq analysis showed that colon cancer marker genes were significantly differentially expressed in the organoids. Specifically colon cancer marker genes such as ANXA1, FABP6, ACE2, FXYD5, LY6E, SERPINE2, SCD, BMP4, CEACAM6, TESC, and TGFBI were overexpressed in the organoids

In drug testing experiments, the organoids used are too small. Some are only 30–40 microns in diameter and contain few cells. The statistical analysis of drug efficacy results seemed not be conducted with the correct number of cells.

Organoids were seeded equally in 384-well plate and the drug sensitivity was measured by the CellTiter-Glo[®] Luminescent Cell Viability Assay whereas images were taken using brightfield microscope at certain magnification.

The quantity of organoids used in experiments differs from the quantity of controls depicted in the mCRC images in the rebuttal letter.

We have provided all the raw images we took for all three organoids.

2. In Figure 2b, the images of PEV and SN38 are cropped from the same image. It appears to be manipulated, for which the authors must provide a convincing explanation.

Figure2a's data is therefore not supported by the images.

The reviewer suspects the remaining data in figure2a (PDOs #2 and #3). The authors must submit the images used to produce the graphs.

Thank you very much for your careful examination of our figures. The experiments were performed and the images were acquired by two individual experimentalists (Dr. Suresh Kumar

and Dr. Yasuhiro Arakawa) who independently generated multiple images from the experiments. This mistake was made accidentally, and we feel extremely sorry for making such a mistake. The PEV image was replaced by another image (technical duplicate). We have also uploaded all the raw images, and the organoids will always be available upon request if any lab is interested to confirm the reproducibility of the experiments. Since the synergy between TOP1 inhibitors and pevonedistat has been validated in cell lines, organoids and mice by multiple experimentalists from different labs, we are confident with the reproducibility of our data.

3. The authors explained that the high concentration of PEV in sample #3 was due to the high CUL4 expression.

The levels of CUL4 expression in #1 and #2 are comparable (Supplementary Figure 4), but their viability is very different. These distinctions are also observed in experiments with cell lines (Figure 1e).

In addition to CUL4, it appears that mCRC 3 must process PEV in high concentration for another reason.

There are a number of molecular factors regulating the sensitivity of the cell to SN38 in addition to the CUL4 complex, which only constitutes a small fraction of repair mechanisms for TOP1 inhibitor-mediated DNA damage (PMID: 27649880, PMID: 32200233, PMID: 32674013).

The graphs in figure 1e exhibit large variations in the y-axis, which infers the diversity of the cell lines. It would be desirable for WB to display the amount of CUL4 expression in these lines.

Do HCT15, KM12, and SW48 express CUL4 at a higher level than 837? SW837 has a vastly superior response.

Comparing the values in Supplemental Figure 1a and Figure 1e, CUL4 expression in HT29 was quite high, and the results were comparable to those in Figure 1e for the cell lines shown. If all the cell lines depicted in Figure 1e have high CUL4 expression levels, the high CUL4 expression in mCRC#3 for the requirement of high C PEV could be supported. If not, another cause may account for the high concentration.

We totally agree with your point, we have performed WB for all these cell lines (see figure blow). We found that SW837 exhibited relative low expression of CUL4B. Again, we believe that there are many molecular factors mitigating the sensitivity of the cell to SN38 other than the CUL4 complex.

additional issues;

1. There are no loading control (tubulin) data in supplementary figure 4. It is necessary to have a complete blot image of the expression of CUL4 and TOP1 with the same amount quantified.

Thank you. As suggested, we have included the loading control for supplementary figure 4c. All uncropped images have been uploaded. Additional data are readily available upon request.

2. In supplementary Figure 4g & h, the authors must illustrate the difference in SN38 reactivity between lines with high and low CUL4 expression, as well as the combination with PEV.

The manuscript's focus lies within the mechanism by which CRC cells repair TOP1-DPC using the neddylation-CUL4 pathway, which is supported by our translational data in CRC models.

Additional experiments in other types of cancers are beyond the scope of the current report.

3. Please check the attached pdf file. Some data were unnecessarily cropped. The reviewer asked the authors to submit original images.

Thank you. Figure 7d uncropped image of DCAF13-FLAG mutants is shown below.

The DNA blots had to be cut and stitched since the slot-blot only has 4 well per row (please see the picture of the apparatus):

Again, we have uploaded all the raw images and all the reagents are available upon request.

Reviewer #4 (Remarks to the Author):

All of my comments and have been addressed in the revised manuscript and I support the publication of this study.

Thank you.

Reviewer #5 - Previously Reviewer #2) (Remarks to the Author):

The authors have addressed my major concerns appropriately. The novelty and quality of the (new) data and the newly provided clarification of major issues have increased the overall quality of the manuscript

substantially. Figure quality and layout also have been improved compared to the first manuscript version.

I recommend the revised manuscript by Sun et al. for publication in Nature Communications.

Thank you.

Reviewers' Comments:

Reviewer #3:

Remarks to the Author:

I read the author's response carefully. I also examined each uploaded organoid images.

However, based solely on the organoid images, I am unable to confirm that the experiments were conducted in good faith. Numerous bands and images are present in the manuscript, which can be easily trimmed and mislabeled. The file name of the uploaded images of organoids can be re-labeled. Even with the raw data, I can only confirm that CRC1 and CRC2 images were created in September 2021, whereas CRC3 images were created in May 2022. I cannot find the clue that the authors used the cropped images only by mistakes. The error in Figure 2b destroyed the trust between the reviewer and the authors.

I am convinced that the author's discovery is incredibly intriguing and original. If I have full confidence in their images and graphs, then the manuscript is entirely novel. All my concerns (except of the duplicated images) were fully addressed by the authors.

I intend to request the editor's final decision on this manuscript. I also instructed the authors not to repeat this mistake in the future.

Reviewer #3 (Remarks to the Author):

Thank you very much for your careful examination of our figures. Again, the experiments were performed and the images were acquired by two individual experimentalists (Dr. Suresh Kumar and Dr. Yasuhiro Arakawa) who independently generated multiple images from the experiments. It is impossible that we duplicated this specific image from a great number of images we took on purpose using a brightfield microscope. This was accidental. Also, we took images of CRC organoids #3 in May 2022 in case the reviewers request. CellTiter-Glo Luminescent Cell Viability assay in Figure 2a is the only method to quantify organoid viability whereas images of the CRC organoid #1 in Figure 2b are complementary to Figure 2a. We will deposit the organoids to NCI Patient-Derived Models Repository once the manuscript is published, so any lab can request them to confirm the reproducibility of our experiments.